# Disulfide-constrained peptide scaffolds enable a robust peptide-therapeutic discovery platform

Lijuan Zhou[1], Fei Cai[1], Yanjie Li[2], Xinxin Gao[2], Yuehua Wei[1], Anna Fedorova[1], Daniel Kirchhofer[1], Rami N. Hannoush[3]*, Yingnan Zhang[1]*

1 Departments of Biological Chemistry, Genentech, Inc., South San Francisco, California, United States of America, 2 Department of Peptide Therapeutics, Genentech, Inc., South San Francisco, California, United States of America, 3 Department of Early Discovery Biochemistry, Genentech, Inc., South San Francisco, California, United States of America

* ramihannoush@gmail.com (RNH); zhang.yingnan@gene.com (YZ)

**Data Availability Statement:** All relevant data are within the manuscript and its Supporting Information files.

## Abstract

Peptides present an alternative modality to immunoglobulin domains or small molecules for developing therapeutics to either agonize or antagonize cellular pathways associated with diseases. However, peptides often suffer from poor chemical and physical stability, limiting their therapeutic potential. Disulfide-constrained peptides (DCP) are naturally occurring and possess numerous desirable properties, such as high stability, that qualify them as drug-like scaffolds for peptide therapeutics. DCPs contain loop regions protruding from the core of the molecule that are amenable to peptide engineering via direct evolution by use of phage display technology. In this study, we have established a robust platform for the discovery of peptide therapeutics using various DCPs as scaffolds. We created diverse libraries comprising seven different DCP scaffolds, resulting in an overall diversity of $2 \times 10^{11}$. The effectiveness of this platform for functional hit discovery has been extensively evaluated, demonstrating a hit rate comparable to that of synthetic antibody libraries. By utilizing chemically synthesized and *in vitro* folded peptides derived from selections of phage displayed DCP libraries, we have successfully generated functional inhibitors targeting the HtrA1 protease. Through affinity maturation strategies, we have transformed initially weak binders against Notch2 with micromolar Kd values to high-affinity ligands in the nanomolar range. This process highlights a viable hit-to-lead progression. Overall, our platform holds significant potential to greatly enhance the discovery of peptide therapeutics.

## Introduction

Many natural peptides have been identified as important regulators of biological functions, acting as hormones, ion channels and GPCR modulators, growth factors, and neurotransmitters. They are usually selective and high-affinity ligands that bind to cell surface receptors and induce intracellular signaling. Therefore, peptides are an attractive modality for developing therapeutics to either agonize or antagonize cellular pathways associated with disease [1]. However, peptides often exhibit poor chemical and physical stability and a short plasma

**Funding:** The author(s) received no specific funding for this work.

**Competing interests:** The authors have declared that no competing interests exist.

circulating half-life, limiting their therapeutic potential [2]. To overcome some of these peptide-intrinsic shortcomings, we considered using naturally occurring and highly stable disulfide constrained peptides (DCPs) as frameworks coupled with display technology aimed at generating de novo binders of therapeutically relevant target proteins [3]. Notably, DCP library strategies has been effectively employed using mRNA display with MCoTI-II as a scaffold [4]. In this study, we employed phage display across seven distinct scaffolds to extent the applicability of this approach.

An important class of DCPs, known as cystine knot peptides (CKP), are naturally found in various plants and animals, exhibiting diverse pharmacological activities [5]. CKPs typically consist of approximately 30–50 amino acids and contain six conserved cysteine residues, which form three disulfide bonds in a I-IV, II-V, III-VI arrangement. Two disulfide bonds form a macrocycle together with the backbone which is penetrated by the third disulfide, resulting in a distinctive knotted topology. It is precisely this disulfide bond-stabilized compact structure that imparts remarkable stability to CKPs, protecting them from thermal, proteolytic, and chemical degradation [6]. The amino acid sequences between the six conserved cysteine residues form multiple surface-exposed loops, which exhibit significant variability in sequence and length. Hence, these loops appear highly suitable for conducting molecular evolution experiments against novel target proteins, enabling the generation of peptides with desirable physicochemical properties [7].

Cyclotides are small cyclic peptides isolated from plants. They typically comprise the cystine knot topology in addition to a distinct head-to-tail cyclized peptide backbone. These collective attributes result in cyclotides exhibiting intrinsic drug-like physicochemical properties akin to non-backbone cyclized CKPs [8].

Other DCPs, which do not possess the cystine knot topology and may have fewer or more than three disulfide bonds, also exhibit exceptional stability and resistance to thermal, chemical, and enzymatic degradation. Examples of such peptides include potentides [9], theta defensin 1 [10], and SFTI-1 [11]. These constrained peptidic frameworks could also be valuable candidates for utilization as scaffolds in peptide engineering endeavors.

Phage display serves as a powerful tool for protein engineering through molecular evolution. Historically, phage display has achieved widespread success in antibody engineering, leading to the development of numerous biotherapeutics currently available on the market [12]. In contrast, although phage display has been employed in peptide engineering for drug discovery purposes for many years, the inherent challenges associated with peptides as therapeutic agents have limited the number of peptide therapeutics developed using the phage display technology. However, there are some notable examples, such as Ecallantide, a treatment for hereditary angioedema [13]; Romiplostim, for treating immune thrombocytopenic purpura, an autoimmune disease [14]; and Tanzeum (albiglutide), a GLP-1 receptor agonist for glycemic control in adults with type 2 diabetes [15]. The development of a robust phage display platform utilizing peptide scaffolds with appropriate pharmacological properties would be highly desirable for efficient discovery of peptide therapeutics. This study introduces strategies that utilize phage display as the primary platform for the direct evolution of peptides based on DCP scaffolds. The objective of this strategy was to identify high-affinity ligands targeting specific proteins of interest, with the ultimate goal of developing peptide therapeutics.

## Results

### Display of DCP scaffolds on M13 surface

Phage display with filament M13 phage utilizes the secretion pathway of *E. Coli* to translocate the DCPs fused to phage coat protein p8 or p3 to the periplasmic space, where the DCPs are

folded. Initially, we selected one CKP scaffold, namely EETI-II [16], and two cyclotide scaffolds, MCoTI-II [17], and kalata B1 [18], to assess the feasibility of displaying DCPs on the surface of M13 phage. The DCPs, incorporating an N-terminal gD-tag for quantification of display levels, were fused at the C-terminal to the major M13 coat protein p8 or the C-terminal domain of minor coat protein p3 (Fig 1A). Moreover, as both EETI-II and MCoTI-II are potent trypsin inhibitors, we measured phage binding to trypsin-coated plates to verify the correct folding of these phage surface-displayed DCPs. For phage display of the cyclotides MCoTI-II and kalata B1, ring opening was necessary to enable fusion with the phage coat proteins, as detailed in the methods section.

All three DCPs, when fused to either p8 or p3, exhibited high levels of display, as demonstrated by low EC50 values derived from phage ELISA titration curve using plate-coated anti-gD antibody (Fig 1B and 1C). Additionally, phage displaying EETI-II and MCoTI-II showed binding signals to immobilized trypsin, confirming functional folding (Fig 1D and 1E). As anticipated, kalata B1 did not exhibit any binding signal against trypsin. In addition to the native scaffolds, we also tested the display of a mutant derived from the EETI-II scaffold (EETI-mut), incorporating four mutations (M7R:Q11R:G22R:P23G), to investigate the adaptability of the scaffolds. The display level on p3 for EETI-mut was similar to that of the parent (Fig 1B), while the display on p8 seemed to decrease slightly but still maintained a high level of display. By contrast, the binding of EETI-mut to trypsin was significantly disrupted by the mutations (Fig 1D and 1E), indicating the mutations interfere the binding of EETI-II to trypsin.

## Construction of phage displayed DCP libraries

Having established that DCPs can be efficiently displayed on phage, we proceeded with seven naturally occurring DCP scaffolds, including EETI-II, AVR9, circulin-A, Conotoxin-MVIIA, Huwentoxin, charybdotoxin and cellulose binding domain (CBD) to construct libraries with diversity in surface-exposed loops as follows (Fig 2).

EETI-II, is a CKP-comprising squash trypsin inhibitor from the seeds of *Ecballium elaterium*, which has been used as a library scaffold for different display methods, including yeast [19] and bacterial surface display [20], as well as mRNA display [21]. For mRNA display, only six residues in Loop 1 (Fig 2A) were randomized to generate a library with diversity of 1 x $10^7$ peptides [21]. For both bacterial and yeast display methods the maximum library sizes are $10^7$–$10^8$, which limits randomization to only one loop of EETI-II, either Loop 1, Loop 2 or Loop 5, per individual library. However, phage display can generate libraries with diversity of up to $10^{11}$, making it possible to simultaneously randomize two loops of EETI-II. Therefore, we designed three EETI-II based libraries in which two loops were randomized simultaneously as well as individually (Fig 2A and 2B). Library I included residue diversity as well as loop length (6, 8, 10 residues) of Loop 1; library II had diversity in Loop 5 residues, but retained the native loop length; library III had diversity in both Loop 1 (residues and length) and Loop 5 (residues only). Altogether, the three libraries contained 1 x $10^{11}$ unique members.

AVR9 [22] is a CKP from the fungus *cladosporium fulvum*. While the structure of this scaffold is currently unavailable, a homology model predicted by DGII [23] exists. According to this model, all loops restrained by disulfide bonds exhibit well-defined secondary structures and are crucial for proper folding, making them unsuitable for randomization. The predicted structure of AVR9 bears resemblance to cyclotide kalata B1 in terms of ring size and the close proximity of its N- and C-termini, which facilitates head-to-tail cyclization. To create a library resembling the cyclotide structure, we modified the design by breaking one loop of the ring (Arg13-Ala14) and fusing the new C-terminus of AVR9 to p8. This arrangement allowed us to

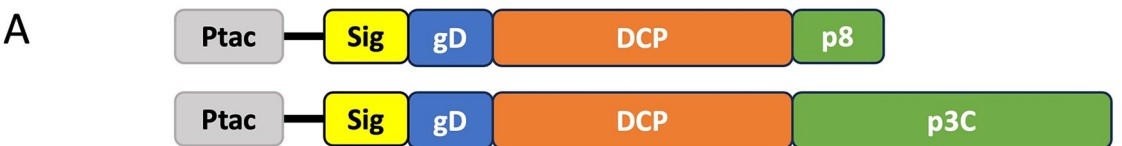

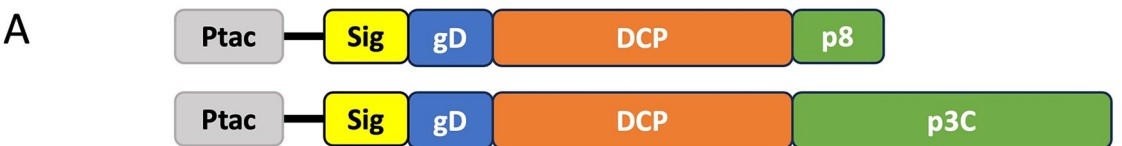

**Fig 1. Phage display of four DCPs, EETI-II, McoTI-II, KalataB1 and EET-mut.** A. Diagram of the DNA constructs for both p3 and p8 phage display, denoting Ptac promoter, secretion signal peptide (Sig), gD tag, DCP and M13 major coat protein p8 or C-term domain of minor coat protein p3 (p3C). The sequences for signal peptide and gD tag are listed. B. Display level on p3 based on detection of gD tag; C. Phage binding to Trypsin with DCP displayed on p3; D. Display level on p8 based on detection of gD tag; E. Phage binding to Trypsin with DCP displayed on p8.

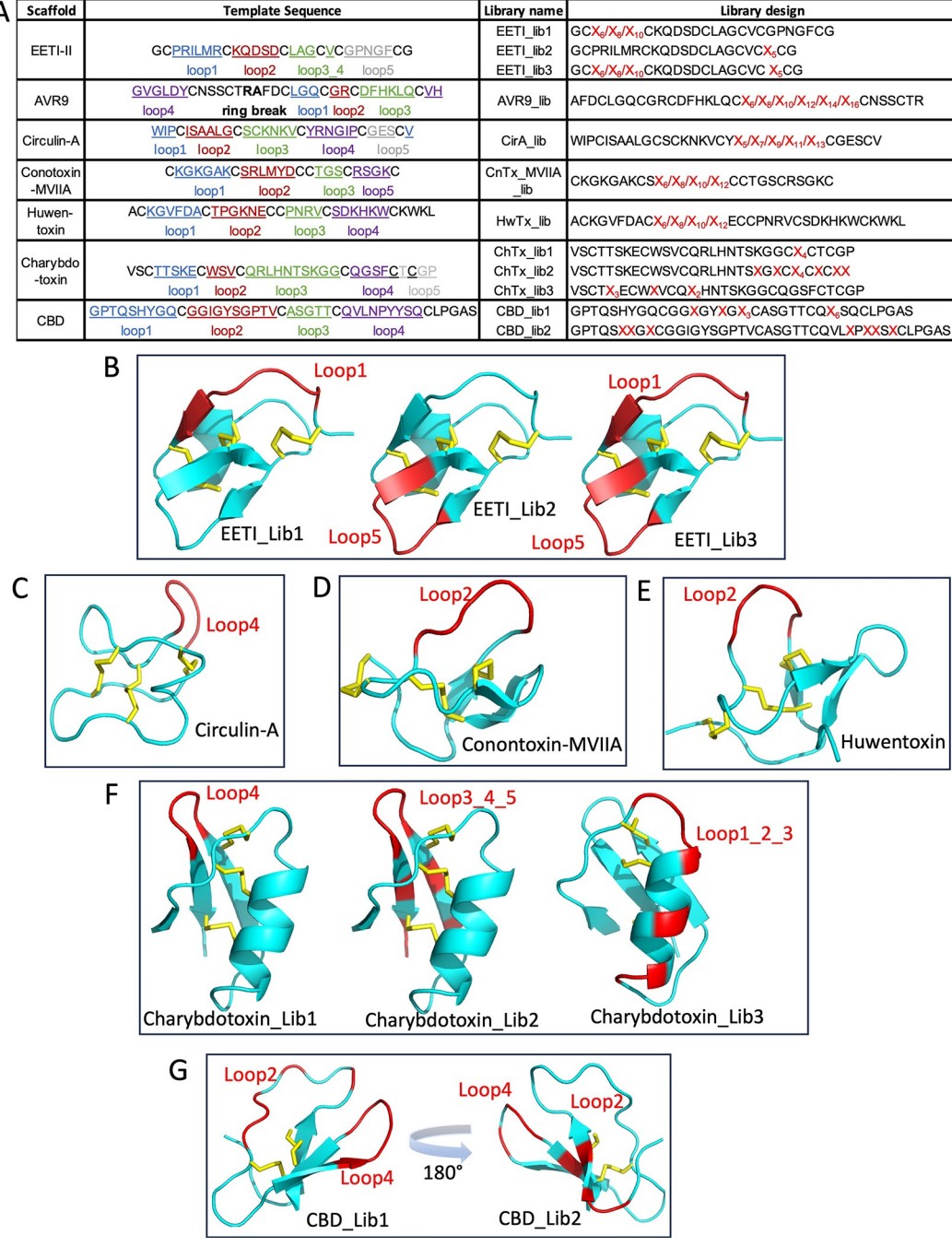

| Scaffold | Template Sequence | Library name | Library design |
|---|---|---|---|
| EETI-II | GCPRILMRCKQDSDCLAGCVCGPNGFCG<br>loop1　　loop2　loop3_4　loop5 | EETI_lib1<br>EETI_lib2<br>EETI_lib3 | GCX₆/X₈/X₁₀CKQDSDCLAGCVCGPNGFCG<br>GCPRILMRCKQDSDCLAGCVCX₅CG<br>GCX₆/X₈/X₁₀CKQDSDCLAGCVC X₅CG |
| AVR9 | GVGLDYCNSSCTRAFDCLGQCGRCDFHKLQCVH<br>loop4　　ring break loop1 loop2　loop3 | AVR9_lib | AFDCLGQCGRCDFHKLQCX₆/X₈/X₁₀/X₁₂/X₁₄/X₁₆CNSSCTR |
| Circulin-A | WIPCISAALGCSCKNKVCYRNGIPCGESCV<br>loop1　loop2　　loop3　　loop4　loop5 | CirA_lib | WIPCISAALGCSCKNKVCYX₅/X₇/X₉/X₁₁/X₁₃CGESCV |
| Conotoxin-MVIIA | CKGKGAKCSRLMYDCCTGSCRSGKC<br>loop1　　loop2　loop3 loop5 | CnTx_MVIIA_lib | CKGKGAKCSX₆/X₈/X₁₀/X₁₂CCTGSCRSGKC |
| Huwen-toxin | ACKGVFDACTPGKNECCPNRVCSDKHKWCKWKL<br>loop1　　loop2　　loop3　　loop4 | HwTx_lib | ACKGVFDACX₆/X₈/X₁₀/X₁₂ECCPNRVCSDKHKWCKWKL |
| Charybdo-toxin | VSCTTSKECWSVCQRLHNTSKGGCQGSFCTCGP<br>loop1　loop2　　loop3　　loop4 loop5 | ChTx_lib1<br>ChTx_lib2<br>ChTx_lib3 | VSCTTSKECWSVCQRLHNTSKGGCX₄CTCGP<br>VSCTTSKECWSVCQRLHNTSXGXCX₄CXCXX<br>VSCTX₃ECWXVCQX₂HNTSKGGCQGSFCTCGP |
| CBD | GPTQSHYGQCGGIGYSGPTVCASGTTCQVLNPYYSQCLPGAS<br>loop1　　loop2　　loop3　　loop4 | CBD_lib1<br>CBD_lib2 | GPTQSHYGQCGGXGYXGX₃CASGTTCQX₆SQCLPGAS<br>GPTQSXXGXCGGIGYSGPTVCASGTTCQVLXPXXSXCLPGAS |

**Fig 2. Sequence and structural representation of library designs for DCP scaffolds.** A. Sequence representation and loop denotation of library design for 7 DCP scaffolds. The loops are color-coded as: blue for loop1, brown for loop2, green for loop3 (or loop3_4) and gray for loop5. All loops' residues are underlined. The ring break point for AVR9 is indicated with bold font. Randomized residues are represented by red font X, representing 19 amino acids excluding Cysteine and the subscripted number of X indicated the length of the randomized region. Structural representation for 6 scaffolds: B. EETI-II (1mro); C. Circulin-A (1bh4); D. Conontoxin_MVIIA (1mvi); E. Humentoxin (1qk6); F. Charybdotoxin (1crd); G. CBD (1cbh). PDB codes are in parenthesis following the name of the scaffold. Regions that had been randomized are colored red. Loops involved in randomization are labelled with red font. Disulfides are shown as yellow. No structure is available for AVR9.

synthetically connect the original N- and C-terminal residues that are in close proximity to form a continuous region suitable for randomization. We carried out hard randomization for this region, varying the length of the randomized segment between 6 to 16 residues (as detailed in Fig 2A). The resulting library size reached a theoretical diversity of $2 \times 10^{10}$.

Additional DCPs included were Circulin A [24], a CKP containing cyclotide from the African plant *chassalia parvifolia*; conotoxin-MVIIA [25], a CKP from the marine cone snail *conus magus*, which acts as a highly selective voltage-gated calcium channel blocker ($Ca_V2.2$), and Huwentoxin [26], a CKP isolated from the Chinese bird spider *Haplopelma schmidti* venom, a nicotinic acetylcholine receptor blocker. Available NMR structures show that these three DCPs each possess a long protruding loop that appear flexible and may tolerate randomization. Therefore, we selected Loop 4 of Circulin A, Loop 2 of conotoxin MVIIA, and Loop 1 for Huwentoxin for hard randomization including loops of various lengths (Fig 2A and 2C–2E). The library sizes were about $2 \times 10^{10}$ for each.

Charybdotoxin [27,28] is a CKP from scorpion toxin that acts as a potassium channel blocker. A phage-displayed library with four randomized residues in a β-turn of charybdotoxin-Loop 4 has been successfully used to identify weak binders of CD4 [28]. We recapitulated this library design (lib 1) and, in addition, constructed two more libraries by extending the randomization to cover more regions on the β-sheet (lib 2), or on the surface of the helix (lib 3) (Fig 2A and 2F). The total diversity of the three libraries combined was $8 \times 10^{10}$.

C-terminal cellulose binding domain (CBD) of cellobiohydrolase I from the fungus *Trichoderma reesei* is a well-characterized DCP scaffold, comprising only two disulfide bonds, that can be used as a prototype for phage display. In previous studies, CBD libraries were constructed in which the relatively flat cellulose binding region of CBD [29], or mainly the CBD-Loop 4 (Fig 2A) were randomized [30], yielding binders with new properties and selectivity against different target proteins. We recapitulated these designs (Fig 2A and 2G) to generate phage-displayed libraries of much higher diversity ($2 \times 10^{10}$) as compared to previously published studies of $5.5 \times 10^8$ and $\sim 4 \times 10^7$.

Together these libraries comprise a large variety of shape and topology. The libraries including EETI-II, conotoxin-MVIIA, Circulin-A, and Huwentoxin as scaffolds were focused on loop randomization. The libraries with CBD as scaffold were more focused on surface randomization, whereas the libraries with charybdotoxin were designed by randomization of particular regions on secondary structures. The total diversity of the ensemble is around $2 \times 10^{11}$.

## Performance of DCP libraries against diverse protein targets

To evaluate the performance of these DCP libraries, we chose a panel of 16 highly diverse proteins for panning trials (Table 1). These soluble proteins cover a wide mass range (11–100 kDa) and have no sequence or structural homology. We employed the DCP libraries listed in Table 1 and successfully retrieved positive hits for 15 out of 16 protein targets (Table 1, S1 Table). In order to capture low affinity binders in the primary panning of all the libraries, we took advantage of multi-valent display by fusing the DCPs to the M13 major coat protein p8 [31]. Positive hits were defined as clones with intact sequences containing all cysteines of their respective DCP scaffold, having spot phage ELISA signals above 0.3 and signal/noise ratios of more than 2. The performances of the DCP libraries varied significantly based on the particular target protein, with the EETI-II libraries being the most successful, yielding 242 unique clones that bound to 14 of 16 protein targets (87.5%). The second-best libraries were based on the CBD scaffold with a success rate of 11/16 protein targets (68.7%) and a total of 172 unique clones. By contrast, the Circulin A and Charybdotoxin libraries had the lowest hit rate of 1/16 protein targets (6.3%) and yielded only 15 and two unique sequences, respectively. The

**Table 1. Total number of unique clones obtained from each DCP scaffold against target proteins.**

|  | EETI-II | AVR9 | Circulin-A | Conontoxin-MVIIA | Huwen-toxin | Charybdo-toxin | CBD |
|---|---|---|---|---|---|---|---|
| HGFA | 6 |  |  |  |  |  | 1 |
| HtrA1 | 6 | 18 | 15 | 6 | 1 |  | 4 |
| ApoL1 | 47 |  |  |  |  |  | 1 |
| R-spondin | 2 |  |  |  |  |  |  |
| Insulin |  |  |  |  |  |  |  |
| Her3-Fc | 11 | 1 |  |  |  | 2 | 1 |
| Lgr5ECD-Fc | 26 |  |  | 2 |  |  | 18 |
| Fz7CRD-Fc | 7 | 1 |  | 1 |  |  | 11 |
| PCSK9 | 20 |  |  | 12 |  |  | 28 |
| PCSK9 ΔCRD |  | 14 |  | 3 |  |  | 2 |
| BamA-amphipol | 1 |  |  |  |  |  |  |
| CD28-Fc | 16 |  |  |  | 4 |  |  |
| CD16A | 44 |  |  |  | 4 |  | 11 |
| EpCAM | 39 |  |  |  |  |  | 90 |
| Notch2Long | 8 |  |  |  |  |  |  |
| Notch2NRR | 9 | 9 |  |  |  |  | 5 |

libraries of the scaffolds AVR9, conotoxin-MVIIA and Huwentoxin performed moderately well, with hit rates of 5/16 (31.2%), 5/16 (31.2%) and 3/16 (18.8%) protein targets, respectively, and the retrieved unique sequences ranged from 9–43. In conclusion, the overall success rate of our DCP library ensemble for 16 protein targets was 93.8%, which is comparable to that of phage-displayed synthetic antibody libraries [32].

## Biochemical characterization of synthesized initial hits

Based on the positive phage ELISA signal, we selected a panel of 34 clones for chemical synthesis and *in vitro* folding to validate ligand target binding. This panel of synthetic peptides represent all seven scaffolds and they cover a wide range of ELISA signals (signal range 0.3–4.0 and signal/noise range 3.0–40). The peptides were generated by solid phase peptide synthesis and folded under oxidative conditions [33]. The formation of disulfide bonds in the final product was confirmed with liquid chromatography-mass spectrometry (LC-MS) (S1 File). Binding of the 24 out of 34 synthetic peptides against their respective protein targets were confirmed by surface plasmon resonance (SPR) measurement (Table 2). Representative binding kinetics of DCP from each of the seven scaffolds are shown in Fig 3A. The affinities for these initial hits are all weak, with $K_d$ values in the micromolar range and with fast on- and off-rates.

In addition to binding, DCPs can be active in function against biological relevant enzymes. HtrA1 is homotrimeric serine protease, which has been implicated in diseases, such as Alzheimer's disease, osteoarthritis, cancer and age-related macular degeneration [34]. The HtrA1 monomer consists of an N-terminal IGFBP-like module and a Kazal-like module, a central trypsin fold protease domain and a C-terminal PDZ domain [35,36]. The library screen identified HtrA1-binding DCPs representing six different scaffolds (Table 2). Following synthesis of nine clones from these six different scaffolds, their inhibitory potential (Table 2) was assessed in an HtrA1 enzyme assay using the fluorescence-quenched synthetic substrate H2-OPT [35]. We found that ligands from three different scaffolds, AVR9, EETI-II and Circulin-A inhibited HtrA1 activity by more than 50% at a concentration of 20 μM (Fig 3B). The most potent inhibitors were AVR62 and the three Circulin-A-based ligands, showing more than 75% inhibition. Among the inhibitors, the Circulin-A scaffold was most strongly represented and the Circulin-

**Table 2. Affinity summary for selected positive hits.** 34 clones with spot ELISA signal range in 0.3–4.0 and signal/noise (s/n) ratio between 3–40 were selected for synthesis and *in vitro* folding, and the $K_d$ of synthetic peptides were measured by SPR with steady state fitting.

| Target | DCP name | Sequence | Kd(μM) SPR | ELISA | s/n ratio |
|---|---|---|---|---|---|
| | **EETI** | **GCPRILMRCKQDSDCLAGCVCGPNGFCG** | | | |
| Notch2Long | N2L-EET-63 | GCEQSTWWAPCKQDSDCLAGCVCYQRWHCG | n/d | 3.9 | 30.8 |
| Notch2Long | N2L-EET-57 | GCDQGHSSGWQKCKQDSDCLAGCVCWFRWHCG | n/d | 1.7 | 17.1 |
| Notch2Long | N2L-EET-83 | GCEQSTWWAPCKQDSDCLAGCVCYLRWHCG | n/d | 3.9 | 30.8 |
| Notch2Long | N2L-EET-62 | GCGQTTAWEPCKQDSDCLAGCVCFMRWHCG | n/d | 2.8 | 32.6 |
| Notch2Long | N2L-EET-03 | GCGWTLWHCKQDSDCLAGCVCEPIWWCD | 43.7 | 0.3 | 3.1 |
| Notch2Long | N2L-EET-36 | GCEVVMERCKQDSDCLAGCVCWYWTSCG | n/d | 2.2 | 18.3 |
| Notch2Long | N2L-EET-28 | GCKQFGNQWICKQDSDCLAGCVCGPNGFCG | n/d | 1.6 | 13.7 |
| Notch2Long | N2L-EET-29 | GCGWTLAHCKQDSDCLAGCVCQPRWVCG | 102.5 | 0.6 | 4.4 |
| Notch2NRR | N2N-EET-57 | GCRPRWGGWHCKQDSDCLAGCVCGPNGFCG | 132.1 | 3.8 | 30.9 |
| Notch2NRR | N2N-EET-33 | GCQQRRWGGWQQCKQDSDCLAGCVCGPNGFCG | 55.6 | 3.6 | 28.9 |
| Notch2NRR | N2N-EET-45 | GCGQQRWGGWPLCKQDSDCLAGCVCGPNGFCG | 55.9 | 3.5 | 14.9 |
| Notch2NRR | N2N-EET-63 | GCMPRYGGWHCKQDSDCLAGCVCGPNGFCG | 166.1 | 1.5 | 21.3 |
| Notch2NRR | N2N-EET-n3 | GCRPRFKGWTCKQDSDCLAGCVCGPNGFCG | 180.1 | 2.1 | 33.1 |
| Notch2NRR | N2N-EET-n13 | GCGQKRMGGWQWCKQDSDCLAGCVCGPNGFCG | 12.0 | 3.4 | 53.9 |
| HtrA1 | HtrA1_EET76 | GCHRPWWQLWGLCKQDSDCLAGCVCGPNGFCG | 9.5 | 1.0 | 13.7 |
| | **CirA** | **WIPCISAALGCSCKNKVCYRNGIPCGESC** | | | |
| HtrA1 | HtrA1_CirA65 | WIPCISAALGCSCKNKVCYIFSKHLCGESCV | 2.7 | 2.2 | 33.9 |
| HtrA1 | HtrA1_CirA38 | WIPCISAALGCSCKNKVCYFLYQICGESCV | 2.5 | 2.1 | 31.9 |
| | **AVR9** | **AFDCLGQCGRCDFHKLQCVHGVGLDYCNSSCTR** | | | |
| Notch2NRR | N2N-AVR-n6 | AFDCLGQCGRCDFHKLQCVWSNRHSPYCNSSCTR | 0.8 | 3.3 | 39.3 |
| Notch2NRR | N2N-AVR-19 | AFDCLGQCGRCDFHKLQCVPYSWTYKQYCNSSCTR | 9.4 | 2.6 | 35.7 |
| HtrA1 | HtrA1_AVR61 | AFDCLGQCGRCDFHKLQCVMVYFIHMTKPYCNSSCTR | > 20 | 1.7 | 24.4 |
| HtrA1 | HtrA1_AVR62 | AFDCLGQCGRCDFHKLQCVRQIVWRVINLSHYCNSSCTR | > 15 | 1.6 | 19.5 |
| | **HwTx** | **ACKGVFDACTPGKNECCPNRVCSDKHKWCKWKL** | | | |
| CD28-Fc | CD28_Hwtx1 | ACKGVFDACTVFEVTDTGVLECCPNAVCSDKHKWCKWKL | n/d | 1.4 | 9.8 |
| HtrA1 | HtrA1_HwTx69 | ACKGVFDACTWKNWRQSRSRECCPNAVCSDKHKWCKWKL | 3.7 | 0.9 | 9.6 |
| | **CnTx** | **ECKGKGAKCSRLMYDCCTGSCRSGKC** | | | |
| PCSK9DCRD | PCSK9_CnTx1 | ECKGKGAKCSYYMGIDKGYVNYCCTGSCRSGKC | 4.5 | 2.0 | 29.7 |
| PCSK9DCRD | PCSK9_CnTx2 | ECKGKGAKCSFMTRQGVQTWCCTGSCRSGKC | 2.8 | 1.7 | 25.9 |
| HtrA1 | HtrA1_CnTx83 | ECKGKGAKCSIQWSVYPWKVCCTGSCRSGKC | 6.4 | 1.1 | 16.1 |
| Fz7CRD-Fc | Fz7-Fc_CnTx1 | ECKGKGAKCSEYWIPMVGWVCCTGSCRSGKC | n/d | 5.1 | 32.7 |
| | **CBD** | **GPTQSHYGQCGGIGYSGPTVCASGTTCQVLNPYYSQCLPGAS** | | | |
| PCSK9 | PCSK9FL_CBD1 | GPTQSNYGMCGGIGYSGPTVCASGTTCQVLYPTTSRCLPGAS | n/d | 1.1 | 17.7 |
| PCSK9 | PCSK9FL_CBD2 | GPTQSKYGMCGGIGYSGPTVCASGTTCQVLDPYTSQCLPGAS | 3.6 | 1.9 | 26.8 |
| Her3-Fc | Her3-Fc_CBD1 | GPTQSWQGRCGGIGYSGPTVCASGTTCQVLTPYWSECLPGAS | 27.0 | 1.7 | 20.2 |
| HtrA1 | HtrA1_CBDwt46 | GPTQSHYGQCGGWGYTGWYQCASGTTCQGPRSRLSQCLPGAS | n/d | 2.6 | 25.9 |
| HtrA1 | HtrA1_CBDamy86 | GPTQSFWGWCGGIGYVGGRYCSGTTCQRPDAHWSQCLPGAS | 158.0 | 1.6 | 22.8 |
| | **ChTx template** | **VSCTTSKECWSVCQRLHNTSKGGCQGSFCTCGP** | | | |
| Her3-Fc | Her3-Fc_ChTx1 | VSCTTSKECWSVCQRLHNTSWGEQCVTKFCKCEE | > 10 | 1.7 | 22.6 |
| Her3-Fc | Her3-Fc_ChTx2 | VSCTTSKECWSVCQRLHNTSWLGSCMERFCKCVD | 1.5 | 1.6 | 24.3 |

A DCP activities were consistent with their binding affinities determined by SPR (Table 2). Therefore, despite having the lowest overall protein target hit rate, the Circulin-A library yielded the highest number of inhibitory HtrA1 ligands. EETI-II is known to be a potent trypsin inhibitor. We derived a HtrA1 inhibitor from the EETI-II scaffold. On testing

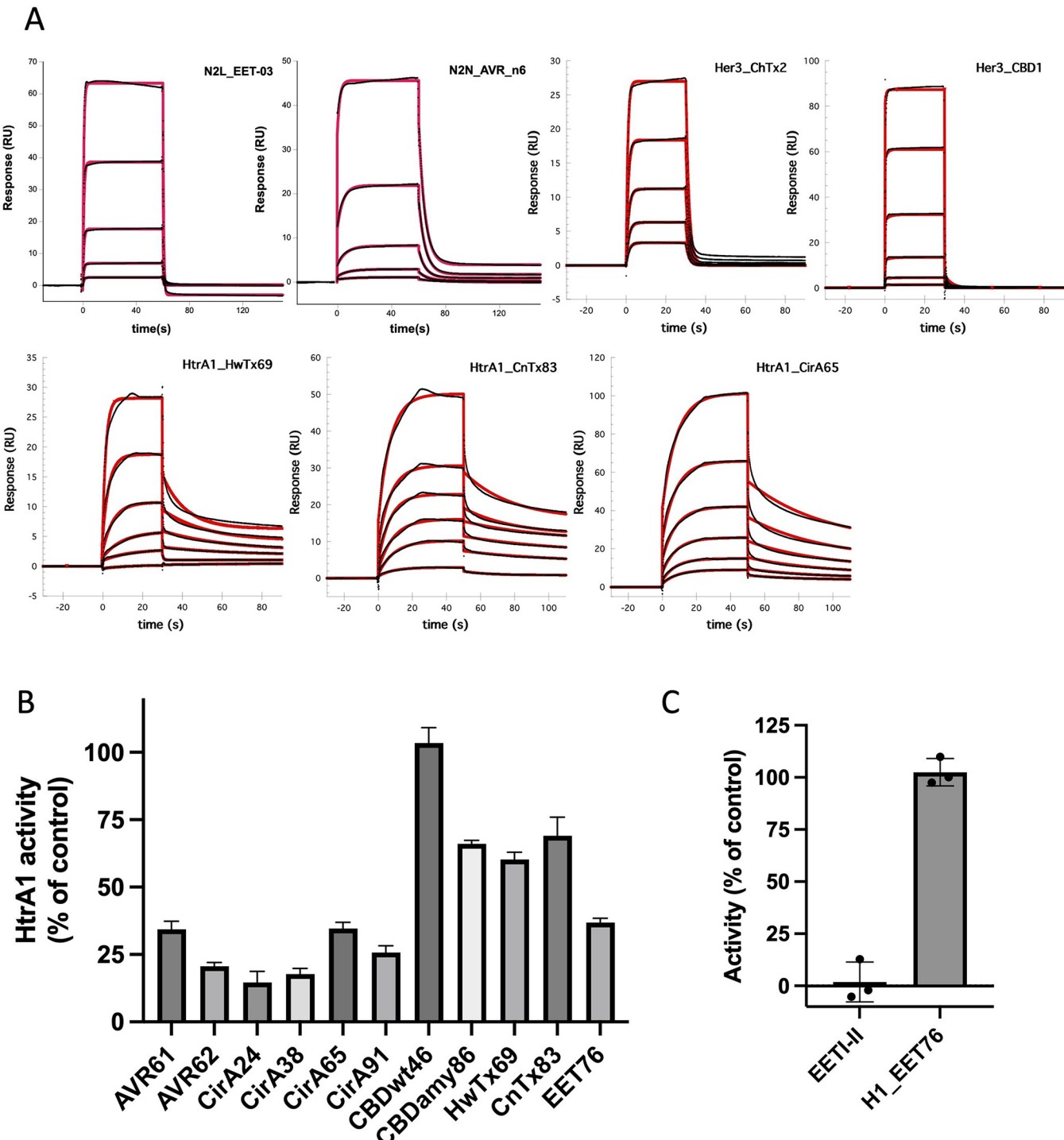

**Fig 3. Biochemical characterization of synthetic DCPs.** A. Representative SPR sensor grams for seven DCP scaffolds. The binding kinetic traces are shown as black and the fitted curve shown as red line. B. Inhibition of HtrA1 enzyme activity by DCPs derived from different scaffolds. C. Inhibition of bovine Trypsin by HtrA1_EET76 vs EETI-II.

HtrA1_EET76's specificity to bovine trypsin, we observed complete trypsin inhibition at 1 μM EETI-II, contrasting with no inhibition by HtrA1_EET76 (Fig 3C).

## Affinity maturation strategies exemplified by anti-Notch 2 ligands

As presented in Table 2, the majority of the hits identified through initial panning against naïve DCP libraries exhibited weak micromolar-range binding to their respective targets. However, for drug leads, it is generally essential to achieve high affinity binding with a Kd in the sub-nanomolar range. Such high-affinity binding can be accomplished through affinity maturation strategies. Therefore, the strongest Notch 2-binding DCP, N2L-EET03 (Kd 43.7 μM, Table 3, S1 File) was selected for directed evolution using a soft randomization strategy (Fig 4A). In this approach, each residue in a given scaffold region has a 50% probability of mutating to any of the possible 20 amino acids, and a 50% probability of remaining as the parental sequence. On the other hand, hard randomization involves a 100% likelihood of mutating any given residue to any of the 20 potential amino acids. We designed a library targeting Loop 1 and Loop 5 (loop designation shown in Fig 2A) simultaneously, which was displayed on the major coat protein p8 and subsequently subjected to panning against Notch 2. During the selection process, an extended washing step was implemented to select for binders with slower off-rates. In order to assess the structure-activity relationship (SAR) of the parent clone, we generated a heat map to evaluate the enrichment score (ES) by comparing the frequency of amino acids at specific positions before and after panning (see methods section).

**Table 3. Affinity summary for clones generated by affinity maturation strategies.** Clones with spot ELISA signal range in 0.3–4.0 and signal/noise (s/n) ratio between 3–40 were selected for synthesis and *in vitro* folding, and the $K_d$ of synthetic peptides were measured by SPR. The two parents' sequences were bolded. The template sequence is shaded with gray.

| DCP name | Sequence | Kd(μM) SPR | ELISA | s/n Ratio |
|---|---|---|---|---|
| **EETI template** | GCPRILMRCKQDSDCLAGCVCGPNGFCG | | | |
| **N2L.EET03** | **GCGWTLWHCKQDSDCLAGCVCEPIWWCG** | 43.7 | 0.4 | 3.1 |
| N2L.EET03.10 | GCGWTLRHCKQDSDCLAGCVCRPIWYCG | 4.5 | 3.0 | 48.4 |
| **N2L.EET31** | **GCIKSHLWCKQDSDCLAGCVCEVWIQCG** | >100 | 0.1 | 0.4 |
| N2L.EET31.32 | GCIKSHLWCPSDERCLAGCVCEVWIQCG | | 0.5 | 3.7 |
| N2L.EET31.43 | GCIKSHLWCKQDSDCEAGCICEVWIQCG | | 0.7 | 5.1 |
| N2L.EET31.43.30 | GCIKSHLWCPHGGRCEAGCICEVWIQCG | | 2.2 | 10.7 |
| N2L.EET31.43.75 | GCIKSHLWCDPRKDCEAGCICEVWIQCG | 0.41 | 2.1 | 10.3 |
| N2L.EET31.43.24 | GCIKSHLWCPKTSDCEAGCICEVWIQCG | 3.5 | 2.1 | 10.5 |
| N2L.EET31.43.35 | GCIKSHLWCPPPRRCEAGCICEVWIQCG | | 2.1 | 11.6 |
| N2L.EET31.43.18 | GCIKSHLWCHPKLDCEAGCICEVWIQCG | | 2.1 | 10.2 |
| N2L.EET31.43.78 | GCIKSHLWCDPRVSCEAGCICEVWIQCG | | 1.9 | 10.3 |
| N2L.EET31.43.88 | GCIKSHLWCDPRKSCEAGCICEVWIQCG | | 1.9 | 10.2 |
| N2L.EET31.43.73 | GCVRSHLWCKQDSDCEAGCICEVWIKCG | | 2.3 | 11.0 |
| N2L.EET31.43.64 | GCIRSHVWCKQDSDCEAGCICEVWVKCG | | 2.0 | 8.8 |
| N2L.EET31.43.43 | GCIRSHVWCKQDSDCEAGCICEIWIQCG | 3.5 | 2.0 | 10.2 |
| N2L.EET31.32.13 | GCIKSHLWCPRDSDCLAGCVCEVWIQCG | 35 | 2.2 | 10.2 |
| N2L.EET31.32.29 | GCIKSHLWCPRHEKCLAGCVCEVWIRCG | | 2.0 | 9.3 |
| N2L.EET31.32.70 | GCIRSHLWCPPSASCLAGCVCEVWIQCG | 1.0 | 1.9 | 7.3 |
| N2L.EET31.32.33 | GCIKSHLWCPRAEKCLAGCICEVWIQCG | | 1.9 | 9.3 |
| N2L.EET31.32.48 | GCIKSHLWCPGKERCLAGCICEIWIQCG | | 1.9 | 7.2 |
| N2L.EET31.32.21 | GCIKSHLWCPSDYCLAGCVCEVWIQCG | | 1.8 | 9.1 |
| N2L.EET31.32.80 | GCIRSQLWCPSDERCLAGCVCEVWIKCG | | 2.1 | 7.6 |
| N2L.EET31.32.85 | GCVRSHVWCPSDERCLAGCVCEIWIRCG | | 2.0 | 4.3 |

The ES heatmap for the N2L-EET03 soft-randomization libraries indicated a preference for the mutation of Trp7 to Arg in Loop 1 and Ile24 to Lys/Arg in Loop 5 (Fig 4B). From this selection campaign, we chose 11 hits for peptide synthesis, and the affinities of the produced DCP were assessed using SPR (S2 Table). The campaign resulted in the creation of the best second-generation binder, N2L.EET03.10, which demonstrated a 10-fold improvement in affinity with $K_d$ value of 4.5 μM (Table 3). Subsequently, we carried out multiple attempts using different randomization strategies to further enhance the affinity; however, none of these efforts yielded any additional improvements.

Subsequently, we selected another primary hit, N2L.EET31, for affinity maturation as outlined in Fig 5A with improved outcome. Despite displaying a weak ELISA signal, N2L.EET31 was identified as the most abundant sequence by NGS analysis (56.6% in round 4). Notably, its enrichment was evident as early as round 2 during the initial panning (Fig 5B). Furthermore, N2L.EET31 exhibited competition with N2L.EET03 in a phage competition ELISA (Fig 5C), suggesting identical or overlapping binding sites. Due to these observations, N2L.EET31 was selected as another candidate for affinity maturation. An NNK-walking experiment (described in the methods section) provided valuable information on the functional importance of individual loop residues. The enrichment score (ES) heatmap of NNK-walking revealed high conservation of Ile3, Ser5, Leu7, and Trp8 in Loop 1, as well as Glu22 and Ile25 in Loop 5 (Fig 5D). These residues are either crucial for binding or play a critical role in overall peptide folding. Additionally, the conservation of Asp14 in Loop 2 and other Loop 2 residues suggested that even though Loop 2 was not randomized in the initial panning, it may be involved in Notch 2 binding. Gly18 in Loop 3 exhibited high conservation, suggesting its structural importance rather than direct involvement in binding. Based on this information, we designed hard randomization libraries for Loop 2 and Loop 3_4 as part of the second generation of affinity maturation.

A total of 58 unique positive clones were identified from the Loop 2 library, while the Loop 3_4 library yielded 82 unique positive clones. Notably, highly conserved Pro10 and Gly18 were observed in the Loop 2 and Loop 3, respectively (Fig 6A). From the second-generation binders, nine top candidates were further cloned in p3 phagemid for ELISA assay. Two representatives displaying positive ELISA signals on p3, namely N2L.EET31.32 and N2L.EET31.43, were selected as parents for the third generation of affinity maturation using libraries displayed on p3 (Fig 6B). In both cases, simultaneous soft-randomization of Loops 1 and 5 was performed. In addition to soft-randomization, N2L.EET31.32 underwent hard-randomization on all residues except the first position of Loop 2, while N2L.EET31.43 was hard-randomized in all residues of Loop 2. A total of 111 and 126 binders were identified for parents N2L.EET31.32 and N2L.EET31.43, respectively. Sequence logo visualization of N2L.EET31.43 (Fig 6C) confirmed the highly conserved residues observed during NNK-walking and second-generation affinity maturation.

Notably, two residues significantly deviated from the parent sequences, Arg4 versus Lys in Loop 1 and Pro11 versus Lys in Loop 2. Such sequence divergence typically indicates that these mutations may contribute to affinity improvement. The majority of binders identified during this campaign exhibited enhanced ELISA signals compared to the parents, indicating significant improvements in affinity (Table 3). Out of the affinity maturation campaign with N2L. EET31, a total of five clones were selected for synthesis, and their $K_d$ values were measured using SPR. Among these clones, N2L.EET31.43.75 emerged as the strongest binder, exhibiting a $K_d$ value of 410 nM—a 250-fold improvement in affinity over the parent DCP.

## Discussion

In this study, we first successfully displayed four DCP scaffolds on the M13 phage surface by utilizing both the major coat protein p8 and the minor coat protein p3. We then demonstrated

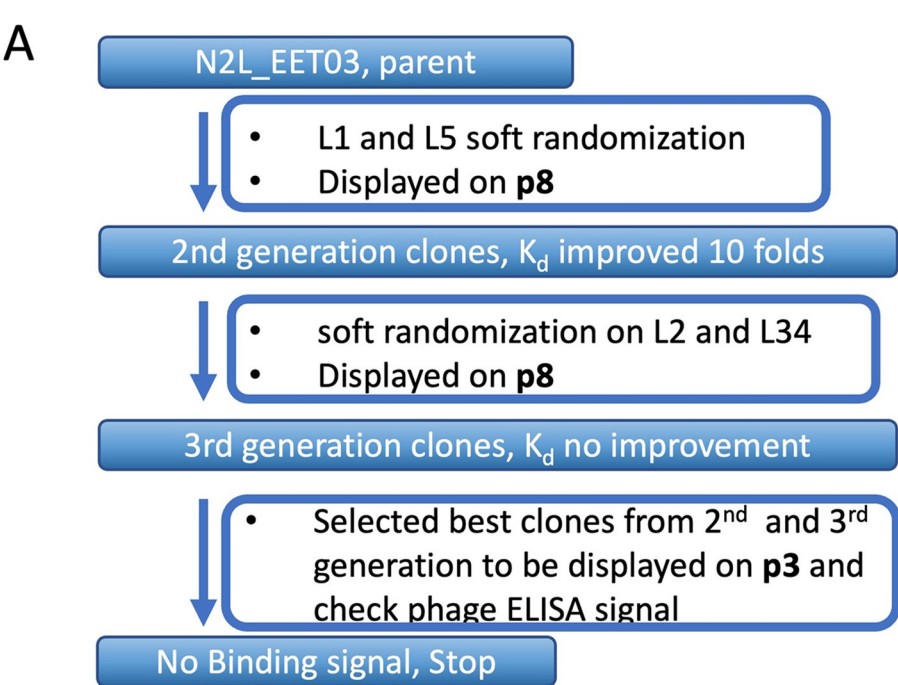

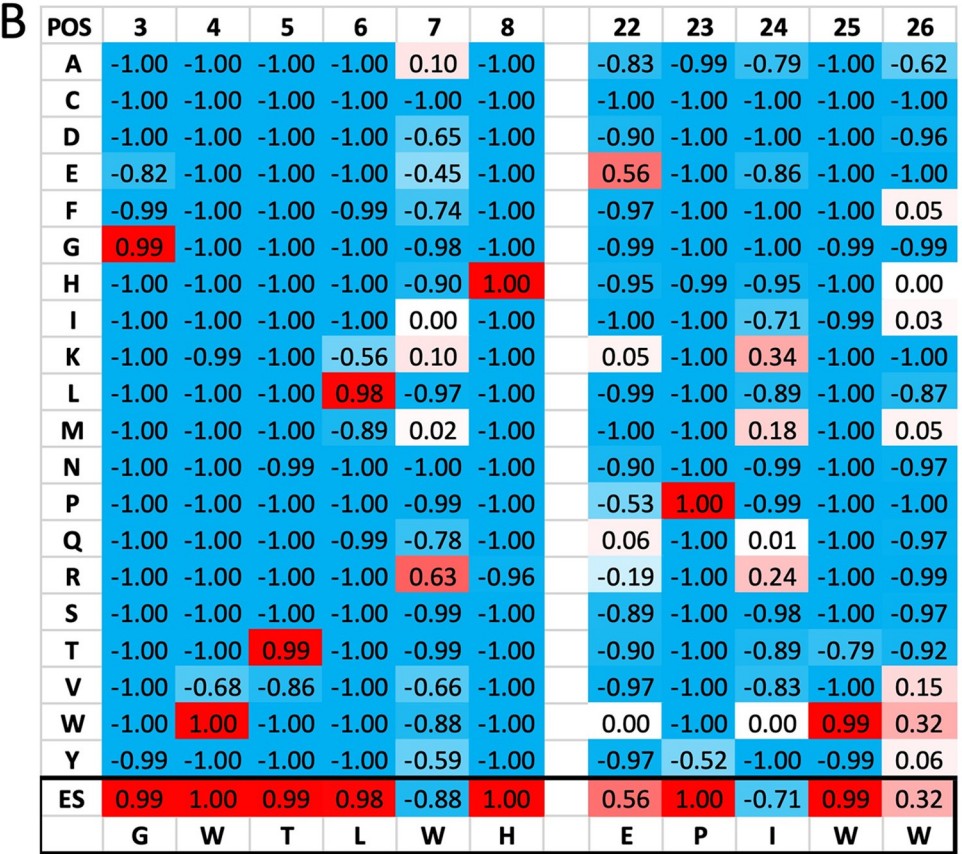

**Fig 4. Affinity maturation of N2L_EET03 with multivalency display.** A. Stepwise affinity-maturation strategy diagram; B. Enrichment Score (ES) heatmap after panning target protein against the soft randomization library of N2L_EET03 displayed on p8.

the binding capabilities of two specific scaffolds, EETI-II and McoTI-II, to their native binding partner trypsin while being displayed on the p8 protein on the phage surface. This finding indicates that these scaffolds are able to retain their correct folding when presented on the phage surface. Kalata B1 is a CKP containing cyclotide. We displayed this cyclotide by breaking one ring and generated peptide with new N- and C-termini, so that it can be fused to phage coat proteins for surface display. The successful display of this cyclotide in the format of DCP opened up the possibility of including cyclotides as DCP scaffolds for library construction. These findings provide evidence for the feasibility of utilizing phage display as a valuable platform for constructing a diverse library of DCP variants for purpose of engineering new functionalities.

We subsequently selected seven DCP scaffolds (including EETI-II) to form the basis of our DCP library ensemble. The design of these libraries was primarily informed by previous reports and insights derived from available DCP structures regarding loop flexibility. However, this approach does not guarantee a reliable design that can maintain native scaffold folding and stability during sequence and loop length randomization. Determining the percentage of DCP library members that effectively fold after being displayed on phage is challenging. Nevertheless, even if only a small fraction of the total library members can achieve the correct fold, we would still have a diverse pool of candidates to select from, provided we attain a library with sufficiently high diversity. In this study, we have achieved a total library size of approximately $10^{11}$, which can withstand a loss of diversity up to two magnitudes due to incorrect folding. Consequently, we believe we would still obtain a reasonably effective library size in the order of $10^9$. Ideally, if we could predict crucial residues within a scaffold that contribute to stability and spontaneous folding and design the libraries by excluding these residues from randomization, we could enhance the percentage of correctly folded members and increase the effective diversity of the resulting library. We anticipate that future library design could benefit from advancements in machine learning technology for protein structure prediction in this regard [37].

The DCP scaffolds we selected in this study were peptides with molecular weight of 3–4 kDa. In contrast to the large surface area available for protein-protein interactions (PPI), including antibody-antigen interactions, where affinities can easily reach the nanomolar range ($K_d$), peptide ligands usually target a much smaller PPI surface area. If the PPI surface area is shallow and flat, such interactions typically exhibit affinities in the micromolar range ($K_d$) [38] Previously, phage display peptide libraries on the major coat protein p8 were developed to capture low-affinity binders by leveraging the avidity effect resulting from multivalent display [31]. We therefore anticipated that initial DCP hits would most likely bind to their targets with low affinity. Hence, we also chose to construct our naïve libraries by displaying DCP on the major coat protein p8 to take advantage of the increased avidity effect. In fact, the majority of hits initially identified during primary panning exhibited affinities in the range of double- to triple- digit micromolar range $K_d$. If we were to use monovalent display with p3, we would most likely be unable to capture these low-affinity binders, resulting in a significant reduction in the libraries' effectiveness.

There are several advantages to our current DCP library ensemble compared to previously reported DCP libraries [19–21,29,30,39]. Firstly, we are able to construct a naïve library with significantly higher diversity, $10^{11}$ compared to a maximum of $10^7$ previously reported.

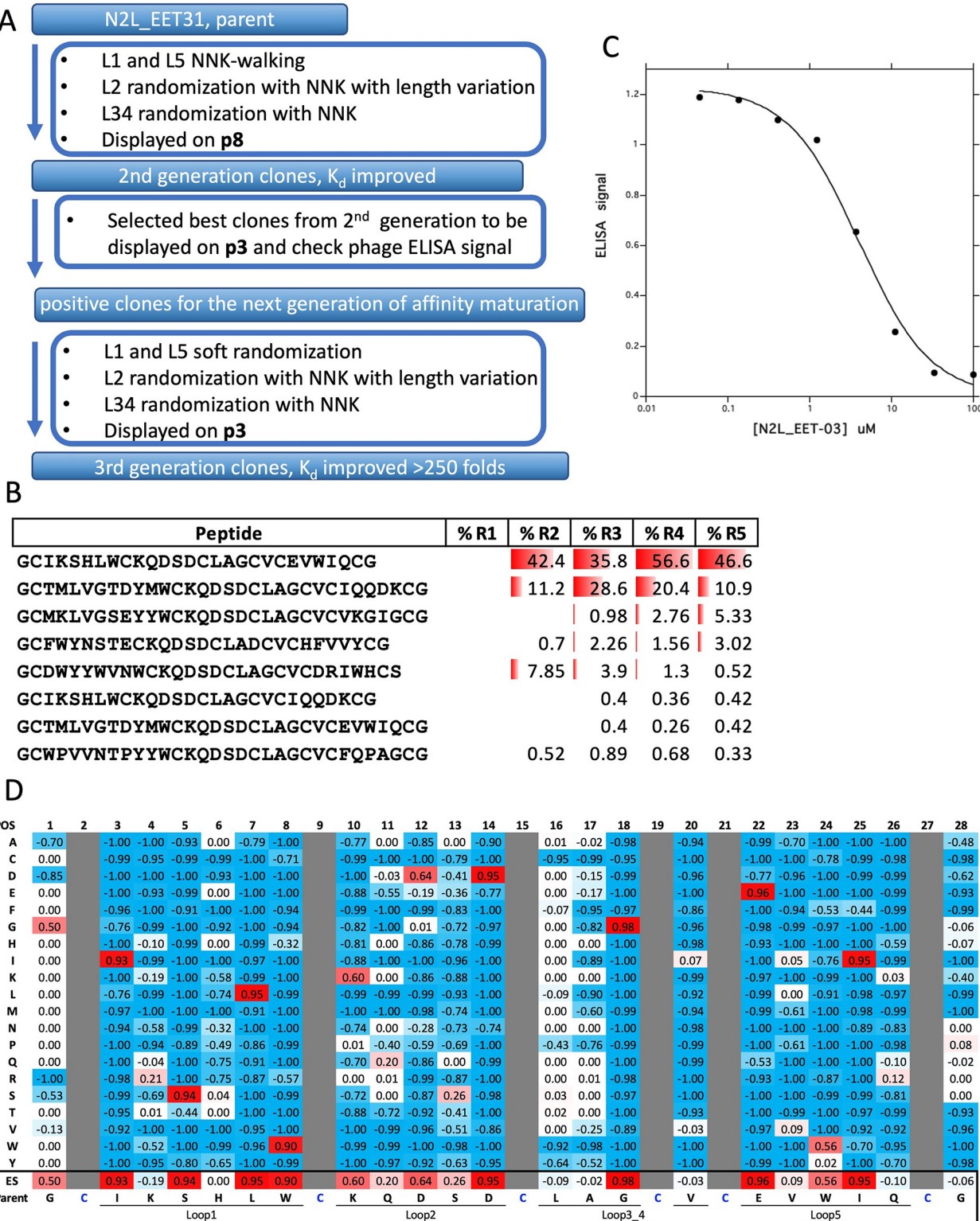

**Fig 5. Affinity maturation of N2L_EET31 with multi- and mono-valency display.** A. Stepwise affinity-maturation strategy diagram; B. N2L_EET31, the top-ranking sequence, identified with NGS; C. N2L_EET03 peptide compete with N2L_EET31 displaying phage in competition phage ELISA; D. Enrichment Score (ES) heatmap after panning target protein against the NNK-walking libraries of N2L_EET31 displayed on p8.

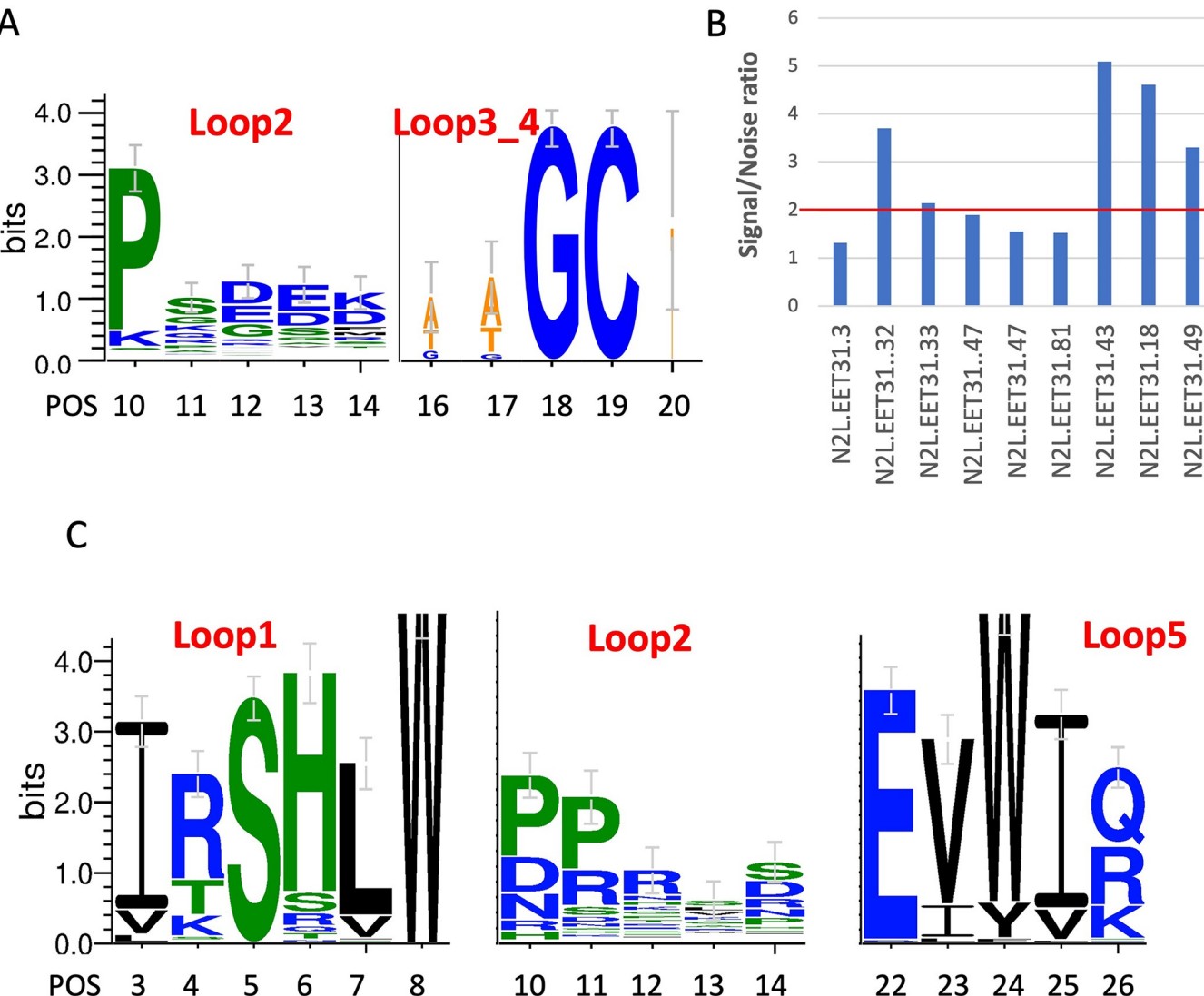

**Fig 6. Affinity maturation of N2L_EET31 with multi- and mono-valency display.** A. Sequence logo after panning against hard randomized libraries of L2 and L3_4 displayed on p8 based on N2L_EET31. B. Screen p8-displayed library derived positive clones on p3 display and identified a few positive clones with signal/noise ratio > 2; C. Sequence logo after panning against soft randomized libraries displayed on p3 based on EET31.43.

Secondly, we have incorporated multiple scaffolds into our library ensemble to enhance the diversity in terms of shape, conformation and secondary structures offered by different scaffolds. As a result, we have achieved a level of library effectiveness from primary panning comparable to that of a synthetic phage-displayed antibody library [32].

The selected positive clones, identified through the phage-displayed library, were produced using solid-phase peptide synthesis and *in vitro* folding [33]. We confirmed that the majority of the positive hits exhibited the expected binding to their target proteins (Table 2). Moreover, we conducted further investigations on the ligands for HtrA1 and identified a few leads that show potential for inhibiting the protease activity of HtrA1 (Fig 3B). These results demonstrate that the sequences of positive hits identified through phage display can successfully be translated into functional molecules through peptide synthesis, followed by *in vitro* folding. The effective production of DCPs through chemical synthesis allows for easy chemical

modification and subsequent incorporation of unnatural amino acids during the lead optimization process to impart drug-like properties of the DCPs.

However, we occasionally observed a discrepancy in binding signals between DCPs displayed on the phage surface and those synthesized *in vitro*. As indicated in Table 2, out of the 34 peptides that exhibited strong binding signals in phage ELISA, only 71% of their synthetic counterparts showed binding to the target proteins, as demonstrated by SPR. This discrepancy suggests that certain DCPs may not fold *in vitro* in the same manner as they do on the phage surface, where the folding of DCPs takes place within the bacterial periplasmic space and is aided by the bacterial secretion machinery throughout the entire process.

Unlike screening with antibody libraries, which typically yield tight binders during primary panning, the primary hits generated from our DCP libraries are predominantly weak binders with $K_d$ values in the micromolar range. In this study, we demonstrated the potential to significantly enhance the affinity of these primary hits through several rounds of affinity maturation campaigns, resulting in the acquisition of high-affinity binders. We presented two distinct examples showcasing affinity maturation strategies.

In the first case (Fig 4), we employed an affinity maturation approach centered around the most potent binder determined by initial ELISA signal ranking. Our strategy involved a stepwise process. Initially, we generated a soft-randomization libraries by displaying the peptides on the major coat protein p8. Subsequently, we would proceed using the best hit from the p8-display library as parent to construct a monovalently displayed soft-randomization libraries to increase the selection stringency. However, panning with p8-displayed library resulted in binders exhibiting only 10-fold improvement in affinity, achieving a $K_d$ in single digit micromolar range. This presented a challenge for the next step as the ligand binding was undetectable in phage ELISA assay under monovalent display conditions with a $K_d$ in micromolar range, rendering further affinity maturation difficult at this point. Consequently, progressing beyond the micromolar range proved unfeasible in this case. Nevertheless, this approach could prove successful if the primary hits or the second-generation hits had already attained nanomolar-range affinity, at which point the monovalent display became detectable in phage ELISA. An illustration of this potential lies in our discovery of potent agonists for the Wnt signaling, wherein the initial hit with a $K_d$ value of 34 nM was improved to become a lead with a $K_d$ value of 1 nM [40].

In the second example (Figs 5 and 6), we started with a phage clone exhibiting extremely low binding affinity, as indicated by a weak ELISA signal and undetectable SPR binding signal. This clone was identified through deep sequencing, which revealed significant enrichment across multiple rounds of bio panning (Fig 5B). Initially, we employed an NNK-walking strategy to gather SAR information for the primary hits. This provided valuable guidance for design of the next generation of affinity maturation libraries. Then, we evaluated a panel of clones from the second generation, displaying them on p3, and identified two clones exhibiting weak ELISA signals. Both clones were chosen as parents for the third-generation affinity maturation. The third round of selection yielded hundreds of binders, with the tightest binder possessing a $K_d$ of 410 nM. By combining deep sequencing, SAR-based library design and monovalent display of libraries, we successfully progressed from primary panning hits with barely detectable affinity ($K_d > 100$ μM) to ligands with $K_d$ values in the nanomolar range. The affinity improvement surpassed 250-fold over three generations of affinity maturation, illustrating the effectiveness of the step-wise evolution strategy.

In conclusion, we have established a robust platform with potential to identify peptide therapeutical if combined with proper strategy of affinity maturation using DCPs as scaffolds. The effectiveness of this platform has been thoroughly evaluated and turned out to be comparable to synthetic antibody libraries. We have successfully demonstrated the feasibility of affinity

maturation transforming weak binders with micromolar $K_d$ values into moderate-affinity ligands in the nanomolar range. However, the process of hit-to-lead for DCP would take much longer time than that for antibody drugs. DCPs possess numerous properties that qualify them as drug-like scaffolds for peptide therapeutics. Our platform, comprising libraries based on a range of DCP scaffolds, has the potential to significantly enhance peptide drug discovery efforts utilizing DCPs as scaffolds in general [40]. While our study focused on only seven scaffolds, it is important to note that there are hundreds of other DCP scaffolds in nature. These additional scaffolds hold potential for inclusion in the library ensemble, further enhancing the robustness of our platform [33].

## Methods

### Protein reagents

Proteins that were purchased commercially are listed in the following in format of protein name (vendor name, catalog number): Her3-Fc (R&D, 348-RB); Fz7CRD-Fc (R&D, 6178-FZ); Insulin (Sigma-Aldrich, I2643) and CD28-Fc (R&D, 342-CD-200). The following human and bacterial (BamA) proteins were produced in house (Genentech) are listed as following in format of protein name (construct, gene ID, expression system): HGF activator (HGFA) (V373-S655, 3083, Hi5 Baculovirus) [41]; HtrA1 (Q23-P480, 5654, Tni Baculovirus) [35]; ApoL1 (D61-L398, 8542, SF9 Baculovirus) [42]; human R-spondin (A32-Q243, 340419, CHO); Lgr5ECD-Fc (M1-W558, 8549, CHO); PCSK9 (M1-Q692, 255738, CHO) [43]; PCSK9ΔCRD (Q31-G452 = PCSK9 lacking the C-terminal Cys-rich domain CRD, 255738, Tni Baculovirus) [44]; BamA-amphipol (BamA protein encapsulated with biotinylated amphipol) (V418-W810, 944870, E. Coli.) [45]; CD16A (G17-Q208, 2214, CHO); hEpCAM (Q24-K265, 4072, CHO); Notch2Long (Notch2 EGF6-EGF12) (D219-E418, 4853, CHO); Notch2NRR (Notch2 NRR domain) (A1423-Q1677, 4853, CHO). Proteins expressed in *Baculovirus* and *E. Coli.* systems were reported previously as referenced. Protein fragments generated by CHO cells were expressed as secreted proteins fused to either Fc (Lgr5ECD-Fc) or His tags (human R-spondin, CD16A, hEpCAM, Notch2Long and Notch2NRR), were purified to >90% purity using affinity chromatography. His tags were subsequently removed via TEV cleavage and the final products were purified with gel-filtration chromatography. Upon purification the proteins, HtrA1, PCSK9FL and PCSK9ΔCRD were *in vitro* biotinylated via Lys conjugation using EZ-Link™ Sulfo-NHS-LC-Biotinylation Kit (ThermoFisher, Cat.21435). CD16A, EpCAM, Notch2Long and Notch2NRR were biotinylated via N-term Avi tag using Enzymatic Protein Biotinylation Kit (Sigma-Aldrich, Cat. CS0008).

### Display of DCPs on M13 phage

Linear sequences of EETI-II (`GCPRILMRCKQDSDCLAGCVCGPNGFCG`), KalataB1 (`GLPVCG ETCVGGTCNTPGCTCSWPVCTRN`), McoTI-II (`SGSDGGVCPKILKKCRRDSDCPGACICRGN GYCG`) and EETI-IImut (`GCPRILKRCKRDSDCLAGCVCRGNGFCG`) was displayed on the surface of M13 bacteriophage by modifying previously described phagemids pS2202d and pS2202b for p3 and p8 display, respectively [46] (S2 and S3 Files). Standard molecular biology techniques were used to replace the fragment of pS2202d encoding gD tag and Erbin PDZ domain with a DNA fragment encoding for these DCPs. *E. Coli* harboring the resulting phagemids were co-infected with M13-KO7 helper phage and cultures were grown in 30 mL 2YT medium supplemented with 50 μg/mL carbenicillin and 25 μg/mL kanamycin at 37˚C for overnight. The propagated phage was purified according to the standard protocol [47] and re-suspended in 1ml PBT buffer (PBS, 0.5% BSA and 0.1% Tween 20), resulting in the production

of phage particles that encapsulated phagemid pS2202d_DCP and displayed DCP. The display level was analyzed using phage ELISA described previously [47].

## Library construction and sorting

The DCP libraries listed in Fig 2A were constructed following Kunkel mutagenesis method [48]. The oligos used for library construction are listed in S3 Table. Residues denoted with "X" were randomized with equimolar of 19 amino acids, except Cys, using trinucleotide oligos. The stop template is the single strand DNA of pS2202b or pS2202d with EETI-II sequence containing three stop codons in both regions of P3–R8 and P23–G25. Twelve libraries based on DCP scaffolds of EETI-II, AVR9, Circulin-A, Conotoxin-MVIIA, Huwentoxin, Charybdo-toxin and CBD were constructed (Fig 2A), each containing ~2 x $10^{10}$ unique members.

The libraries were pooled into two separate bins: three libraries made from EETI-II scaffold were mixed together as one pool and the remaining nine libraries mixed together as the second pool. Two pools of the libraries were cycled through rounds of binding selection either with solution panning protocol [44] against biotinylated HtrA1, PCSK9FL, PCSK9dCRDTEV, BamA-amphipol, CD16A, EpCAM, Notch2Long and Notch2NRR; or plate sorting protocol [47] against HGFA, R-spondin, Insulin, ApoL1, Her3-Fc, Lgr5ECD-Fc, Fz7CRD-Fc and CD28-Fc. After four to five rounds of binding selection, individual phage clones were picked, propagated and purified, and was analyzed with phage ELISA. Positive clones were subjected to either Sanger DNA sequence analysis or next generation sequencing (NGS).

## Phage ELISA

After four rounds of binding selection, individual phage clones were picked and inoculated into 450μl 2YT media containing 50μg/ml Carbenecilin and M13-KO7 helper phage in 96-well blocks, which were grown at 37°C overnight. The supernatant was analyzed with spot phage ELISA as follows: target proteins were captured to 384-well Maxisorp immunoplates either through direct coating or affinity tag on protein, phage supernatant diluted (1:3) with PBT buffer (PBS with 0.5% BSA and 0.05% Tween 20) was added to the wells. The plates were washed and bound phage was detected with anti-M13-HRP followed by TMB substrate. In these assays, phage binding to BSA was tested in parallel to assess background binding. S/N is calculated as the ration between signals for target proteins over BSA (background). S/N > 2 were considered positive. Positive clones were subjected to DNA sequence analysis.

## Peptide synthesis and in vitro folding

DCPs were produced with solid phase peptide synthesis. Briefly, linear peptides were synthesized, purified and folded using either buffer 1 (0.1 M ammonium bicarbonate, pH 9.0, 2 mM reduced glutathione, 0.5 mM oxidized glutathione, 4% DMSO) or buffer 2 (0.1 M ammonium bicarbonate, pH 8.0, 1 mM reduced glutathione, 50% DMSO) at 0.5 mg/mL for 24 h at RT with shaking. The folded peptides were purified with a C18 reversed phase HPLC column.

The final products were confirmed with LC-MS [33]. Peptide content was calculated using amino acid analysis. Selected peptides were synthesized by CS Bio (Menlo Park, CA).

## NNK-walking

A top hit (N2L.31) against target hNotch2 was identified from the initial panning using the EETI-II library. For each non-cysteine position, an oligomer was designed with a NNK codon for mutagenesis at corresponding peptide position. Total 28 NNK-walking oligomers were synthesized, each with a 15-nucleotides flanking regions that are complementary to the

phagemid at both ends. Oligomers were pooled at equal molar ratio, then the pooled oligomer was used to construct NNK-walking library, using the standard Kunkel mutagenesis method [48]. This NNK-walking library was panned against biotinylated hNotch2 in solution using M-280 Streptavidin Dynabeads (Invitrogen 11205D) first, then followed by one round of solid-phase panning using Neutravidin-coated NUNC MaxiSorp strips (Thermo 95029140). Solution and solid-phase panning used 200 nM and 20 nM biotinylated hNotch2, respectively. Phage samples were kept at each step for NGS analysis later.

## Affinity maturation

Based on NNK-walking results, one soft- and two hard-randomization libraries were designed for affinity maturation, targeting Loop 1 and Loop 5, and Loops 2, and 3 and 4, respectively. For soft randomization of Loops 1 and 5, doping codon, designated as a 70:10:10:10 mixture of nucleotide bases and the wild type base was in excess, was used for each amino acid position in the targeting region and this results in approximately 50% mutation rate at the amino acid level. For hard-randomization of Loops 2, and 3 and 4, a NNK codon was used for each targeting position. These three libraries were each panned against biotinylated Notch 2 with one round of solution panning using M-280 Streptavidin Dynabeads followed by three rounds of solid-phase panning using MaxiSorp strips coated with Neutravidin or Streptavidin, alternatively between rounds.

Top hits identified from the above affinity maturation effort were cloned into p3 phagemid for ELISA. Two ELISA positive ones, namely N2L.31.L2.32 and N2L.31.L34.43, were picked for further improvement. In each case, one soft- and one hard-randomization libraries were designed targeting loops 1 and 5 and loop 2, respectively, using strategies described above. For each library sorting, one round of solution panning using M-280 Streptavidin Dynabeads followed by four rounds of solid-phase panning using MaxiSorp strips coated with Neutravidin or Streptavidin, alternatively between rounds.

## Next generation sequencing (NGS)

Phage samples were kept at each step, representing a fraction of libraries, elution and negative controls. These samples were subjects to NGS analysis using Illumina MiSeq system with MiSeq Reagent Kits v2. Sample preparation and sequencing was performed by following manufacture's instruction. Briefly, two rounds of PCR were used to amplify each sample. The first PCR adds a unique index to each sample; while the second PCR adds universal overhangs at both ends for sequencing purpose. Secondary PCR products were pooled, column purified and then quantified by qPCR, which is performed using KAPA Library Quant kit (Illuimina KK4835). Finally, NGS sample was run on a MiSeq with the "Generate FASTQ" module.

## Enrichment Score (ES) calculation and ES heatmap generation

The sequences before and after sorting were all in same length. The frequencies for each of 20 amino acids at each position were calculated as fraction of the total count. F1 is the fraction in sorted sample; F0 is the fraction in unsorted sample. The enrichment score (ES) for a specific amino acid at a specific position was calculated as following:

If $F1 > F0$, $ES = (F1-F0)/(1-F0)$

If $F1 < F0$, $ES = (F1-F0)/F0$

Therefore, $-1 < ES < 1$, with negative number indicates de-enrichment and positive number indicates enrichment.

A two-dimensional (2D)-matrix table of ES can be calculated by comparing NGS datasets before and after sorting with sequence position number in x-axis and 20 amino acids on y-

axis. A heatmap was generated by excel based on the 2D-matrix table using gradient 3-color scale, with blue for -1, white for 0 and red for +1.

## Surface plasma resonance (SPR)

All binding experiments were performed using a Biacore S200 instrument (GE Healthcare) at 25˚C at a flow rate of 50 μL/min. The kinetics and affinity measurements were carried out using either single-cycle or multi-cycle kinetics (SCK or MCK), solvent corrected and referenced by subtracting the signal to the blank flow cell. Binding affinity and kinetic parameters (association and dissociation rate constants, and dissociation equilibrium constant) were calculated with the Biacore S200 Evaluation Software (version 1.1.1, Cytiva) using either a 1:1 binding model or a steady-state equilibrium model (according to the manufacturer's instructions). Final kinetics, affinity and standard deviations were calculated based on at least three independent experiments, and the standard deviations were typically less than 10%.

The binding affinities of DCPs and HtrA1 FL were determined with 10 mM HEPES (8.0); 200mM NaCl; 0.25% (v/v) CHAPS; 5% (v/v) DMSO as running buffer. Binding of HtrA1_HwTx69, HtrA1_CnTx83, HtrA1_CirA65 were measured using Series S Sensor Chip SA (Cytiva) captured approximately 900 RUs of biotinylated HtrA1 FL. While the binding affinity of HtrA1_CnTx83 and HtrA1_CirA65 DCPs were analyzed using MCK in a two-fold dilution series starting from 5 μM or 10 μM final concentrations, HtrA1_HwTx69 were studied using MCK in a three-fold dilution series starting from 10 μM.

Her3-Fc Avi-tag (R&D) was captured at about 1200 and 3000 RU on Series S Sensor Chip SA (Cytiva). The SPR measurements were done in HBS-EP+ buffer (0.01 M HEPES pH 7.4, 0.15 M NaCl, 3 mM EDTA, 0.05% v/v Surfactant P20) plus 1% v/v DMSO. The binding affinity of Her3_ChTx2 was analyzed using MCK in a two-fold dilution series starting from 20 μM while Her3_CBD1 was in a three-fold dilution series starting from 100 μM.

Biolytined hPCSK9 FL was captured at about 600 and 800 RU on Series S Sensor Chip SA (Cytiva). The binding affinity of hPCSK9 DCPs to hPCSK9 FL were analyzed using MCK in a three-fold dilution series starting from 10 μM or 50 μM in a running buffer (10 mM HEPES (pH7.4), 150mM NaCl, 0.05% v/v Tween20, 2% v/v DMSO) at a flow rate of 50 μL/min.

rhCD28_Fc (Asn19-Pro152, Cat#342-CD, R&D) were captured at approximately 500, 950 and 1500 RU on Sensor Chip Protein A. CD28-HwTx1 in HBS-EP+ buffer plus 1% v/v DMSO with a three-fold dilution series starting from 100 μM were analyzed. No binding signal was detected using SCK.

Fz7CRD-Fc (Cat# 6178-FZ, R&D) were captured at around 800 and 2000 RU on Sensor Chip Protein A. hFz7-CrTx1 in HBS-EP+ buffer plus 5% v/v DMSO with a three-fold dilution series starting from 100 μM were analyzed. No binding signal were detected using SCK.

Biotinylated N2L protein (D219–E418, made in-house) and NRR protein (A1423–Q1677, made in-house) were captured at around 500, 1000 and 1500 RU on Sensor Chip SA (Cytiva). Binding affinity of N2L and N2N DCPs with their respective protein was measured in the running buffer (50 mM Tris, pH8.0, 100 mM NaCl, 2 mg/mL PEG3350, 1 mg/mL CM dextran, 1 mg/mL BSA, 0.0025% Tween20) with 3-fold serial dilution starting from 100 μM of peptides.

## HtrA1 and bovine trypsin enzyme assay

Human full-length HtrA1 (HtrA1 FL) was expressed in *Trichoplusia ni* cells and purified as described [35,49]. Enzyme assays with the fluorescence-quenched synthetic substrate Mca-IRRVSYSF(Dnp)KK (H2-OPT) [35,50] were carried out essentially as described (Eigenbrot 2012). DCP stock solutions in DMSO were diluted in 50 mM Tris-HCl, pH 8.0, 200 mM NaCl, 0.25% CHAPS (assay buffer) and incubated with HtrA1 in 96-well black optical bottom plates

for 15 min at 37˚C. A 10 mM stock solution of H2-OPT in DMSO was diluted with water, pre-warmed at 37˚C and then added to the DCP-enzyme mixtures. The final concentrations in the reaction mixtures were: 1 nM HtrA1, 20 μM DCP, 5 μM H2-OPT, 0.1% DMSO. Substrate cleavage was measured at 37˚C on a SPECTRAmax M5 microplate reader (Molecular Devices) and the initial rates of cleavage determined. Bovine Trypsin (Sigma T1426) enzyme assays were carried out similar to the above, with buffer of 50mM Tris-HCL pH8.0, 100mM NaCl, 0.1% Triton X-100 with reaction mixture of 1nM Bovine Trypsin, 1 μM EETI-II or HtrA1_EET76, 0.4mM S2222, 0.05% DMSO. The enzyme activities were expressed as percent of control values (without DCP). The determined values represent the average ± S.D. of three independent experiments.

## Supporting information

**S1 Table. Sequences and associated ELISA results for 507 positive clones outlined in Table 1.** The sequences are grouped into spreadsheets with target protein as spreadsheet names. The scaffold sequences and names are in bold font.
(XLSX)

**S2 Table. Affinity summary for clones generated by affinity maturation of N2L.EET03.**
(XLSX)

**S3 Table. List of oligos for constructing libraries.**
(XLSX)

**S1 File. CoA (Certificate of Analysis) for all DCPs listed in Table 2, Table 3 and S2 Table.** A zip file contains 51 pdf files with filenames are the same as the "DCP name" listed in the tables.
(ZIP)

**S2 File. Full sequence of phagemid pS2202b for p8 display.**
(TXT)

**S3 File. Full sequence of phagemid pS2202d for p3 display.**
(TXT)

## Author Contributions

**Conceptualization:** Xinxin Gao, Rami N. Hannoush, Yingnan Zhang.

**Data curation:** Lijuan Zhou, Fei Cai, Yanjie Li, Xinxin Gao, Yuehua Wei, Anna Fedorova, Daniel Kirchhofer, Rami N. Hannoush, Yingnan Zhang.

**Formal analysis:** Lijuan Zhou, Fei Cai, Yanjie Li, Xinxin Gao, Yuehua Wei, Daniel Kirchhofer, Rami N. Hannoush, Yingnan Zhang.

**Methodology:** Lijuan Zhou, Fei Cai, Xinxin Gao, Rami N. Hannoush, Yingnan Zhang.

**Project administration:** Rami N. Hannoush, Yingnan Zhang.

**Resources:** Rami N. Hannoush.

**Supervision:** Rami N. Hannoush, Yingnan Zhang.

**Writing – original draft:** Lijuan Zhou, Fei Cai, Daniel Kirchhofer, Yingnan Zhang.

**Writing – review & editing:** Xinxin Gao, Daniel Kirchhofer, Rami N. Hannoush, Yingnan Zhang.

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
