## [Decision Letter · Decision Letter 0]

21 Nov 2023

PONE-D-23-32053Disulfide-constrained peptide scaffolds enable a robust peptide-therapeutic discovery platformPLOS ONE

Dear Dr. Zhang,

Thank you for submitting your manuscript to PLOS ONE. After careful consideration, we feel that it has merit but does not fully meet PLOS ONE’s publication criteria as it currently stands. Therefore, we invite you to submit a revised version of the manuscript that convincingly addresses all the points raised during the review process. Please submit your revised manuscript by Dec 29 2023 11:59PM. If you will need more time than this to complete your revisions, please reply to this message or contact the journal office at plosone@plos.org. Please include the following items when submitting your revised manuscript:A rebuttal letter that responds to each point raised by the academic editor and reviewer(s). You should upload this letter as a separate file labeled 'Response to Reviewers'.A marked-up copy of your manuscript that highlights changes made to the original version. You should upload this as a separate file labeled 'Revised Manuscript with Track Changes'.An unmarked version of your revised paper without tracked changes. You should upload this as a separate file labeled 'Manuscript'.If applicable, we recommend that you deposit your laboratory protocols in protocols.io to enhance the reproducibility of your results. Protocols.io assigns your protocol its own identifier (DOI) so that it can be cited independently in the future. For instructions see: https://journals.plos.org/plosone/s/submission-guidelines#loc-laboratory-protocols. Additionally, PLOS ONE offers an option for publishing peer-reviewed Lab Protocol articles, which describe protocols hosted on protocols.io. Read more information on sharing protocols at https://plos.org/protocols?utm_medium=editorial-email&utm_source=authorletters&utm_campaign=protocols.

We look forward to receiving your revised manuscript.

Kind regards,

Maria Gasset, Ph.D.

Academic Editor

PLOS ONE

Journal Requirements:

Reviewers' comments:

Reviewer's Responses to Questions

**Comments to the Author**

1. Is the manuscript technically sound, and do the data support the conclusions?

Reviewer #1: Yes

Reviewer #2: Yes

2. Has the statistical analysis been performed appropriately and rigorously? 

Reviewer #1: N/A

Reviewer #2: Yes

3. Have the authors made all data underlying the findings in their manuscript fully available?

Reviewer #1: No

Reviewer #2: Yes

4. Is the manuscript presented in an intelligible fashion and written in standard English?

Reviewer #1: Yes

Reviewer #2: Yes

5. Review Comments to the Author

Reviewer #1: Zhou et al describe the design and application of a number of new phage display libraries based on 7 cysteine constrained peptide scaffolds. They screen these libraries against 16 representative proteins identifying a range of moderate strength peptide binders. Two selected peptides are further affinity matured, in one case resulting in a peptide that binds in the mid-nanomolar range – still relatively weak for a cyclic peptide binder, but approaching an affinity which could be useful as a starting hit. On the whole the manuscript is clearly written and the conclusions presented are fair and valid. However, no supplementary information has been included which means much of the data needs to be taken on trust. Raw data needs adding to an SI before publication. More detailed methods are also required. Other than that I believe the reported data is sound and I have no hesitation recommending its publication following some additional minor modifications which I list below:

- Pg 3 line 40-42: I don’t really understand this sentence. Where do the immunoglobin domains come in? These haven’t been mentioned previously

- similar library approaches have been applied with mRNA display e.g. Liu et al JACS 143 18481 2021 and this work should be mentioned and cited in the introduction.

- pg 5 line 92: what is a gD tag? More generally I think it would be helpful to have a figure illustrating the library design described in lines 92-98.

- pg 5 line 99-100: Fig 1 shows a series of ELISA curves and this sentence states that “high levels of display” were observed. How is this quantified? What is “high” relative to? Can this data be more robustly quantified?

Pg 5 line 102: State “confirming functional folding” however the signal seems much lower here for a given Phage OD (again it would be useful to have actual quantification of this). Can you comment on this? Is it because a large portion of the expressed peptide is not folded correctly?

Pg 13 lines 239-243: Two residues are highlighted based on screening as showing favourable substitutions. In the improved peptide a third substitution has also been made W22Y. Please elaborate further. Were more peptides synthesised and tested and the one currently presented in table 3 is just the best? If so, all of this data should be included.

Pg 18 line 322-323: I’m not sure about this comment. Many PPIs are much tighter than µM. Please provide a reference to back this up.

Pg 21 line 394-5: I think it is too strong a claim to say this is a robust platform for discovery of peptide “therapeutics”. Also be what measure is this platform comparable to synthetic antibody libraries? I would say that the peptides being isolated are somewhat weaker binders than things derived from antibody libraries.

Pg 23 line 442: method for phage ELISA should be included since it is used substantially in the main figures. Please also explain how the S/N is calculated

Pg 24 peptide synthesis: this method should be included since there are many variations and the folding is non standard. All LC-MS data should be included in the SI

Examples of data that should be included in SI:

- Please include all ELISA data. E.g. Pg 9 line 178-180

- Please include all peptide LC-MS

- Please include all primers used for e.g. library construction

- Please include full sequences for phage libraries

Reviewer #2: In this manuscript Zhang et al describe a phage display platform for discovery of peptide therapeutics using diverse disulfide-constrained peptides as scaffolds. The screening platform yielded functional hits against a range of protein targets including HtrA1 and Notch2. A thorough and cleverly designed affinity maturation strategy is also described, which enabled hit-to-lead optimisation of a Notch2 ligand, providing a nanomolar binder. The platform could be highly beneficial for discovering therapeutic peptides and thus this paper will be of interest to the peptide community. I am in favour of publication after the minor corrections listed below are addressed.

Minor Corrections

1. Can you comment on the specificity of any of your hits? For example, EETI-II is a known trypsin inhibitor. Do your EETI-II-based HtrA1 inhibitors also inhibit trypsin? As you are optimising for affinity, it would be good to know that you aren’t isolating promiscuous ligands.

2. I’m sure this will be fixed in the final manuscript, but I found several figures, in particular figure 2, difficult to read due to poor resolution. Please provide clearer figures.

3. Could you include a sentence in the manuscript describing hard and soft randomisation. I think this would be useful for readers not familiar with this concept.

6. PLOS authors have the option to publish the peer review history of their article (what does this mean?). If published, this will include your full peer review and any attached files.

Reviewer #1: No

Reviewer #2: **Yes: **Scott Lovell

---

## [Author Response · Author response to Decision Letter 0]

23 Jan 2024

Dear Editor:

Thank you for your careful review of our manuscript titled “Disulfide-constrained peptide scaffolds enable a robust peptide-therapeutic discovery platform” (PONE-D-23-32053). We appreciated the positive feedbacks and comments from both reviewers and have revised our manuscript accordingly with tracked change. We added two figure panels per reviewer’s requests (Fig 1A and Fig 3C) and six files (S1-S6) for Supporting Information. The titles and the list for Supporting Information are added as a new page in main text after Figure Legend section. 

Following are our responses to two reviewers point by point. The reviewers’ comments/questions are in bold font and our responses are in regular font:

Reviewer #1: Zhou et al describe the design and application of a number of new phage display libraries based on 7 cysteine constrained peptide scaffolds. They screen these libraries against 16 representative proteins identifying a range of moderate strength peptide binders. Two selected peptides are further affinity matured, in one case resulting in a peptide that binds in the mid-nanomolar range – still relatively weak for a cyclic peptide binder, but approaching an affinity which could be useful as a starting hit. On the whole the manuscript is clearly written and the conclusions presented are fair and valid. However, no supplementary information has been included which means much of the data needs to be taken on trust. Raw data needs adding to an SI before publication. More detailed methods are also required. Other than that I believe the reported data is sound and I have no hesitation recommending its publication following some additional minor modifications which I list below:

- Pg 3 line 40-42: I don’t really understand this sentence. Where do the immunoglobin domains come in? These haven’t been mentioned previously

Deleted “beyond immunoglobin domains” 

- similar library approaches have been applied with mRNA display e.g. Liu et al JACS 143 18481 2021 and this work should be mentioned and cited in the introduction.

Add “peptide library approaches have been applied with mRNA display” at line 44 and the reference is included. 

- pg 5 line 92: what is a gD tag? More generally I think it would be helpful to have a figure illustrating the library design described in lines 92-98.

A new panel Figure 1A is added in Figure 1 with diagram of the constructs for both p8 and p3 display and cited in line 97. The sequences for signal peptide and gD tag are also listed in the figure. The references to other Figure 1 panels are modified accordingly in line 103-116 and the legend for Figure1 was modified as well. 

- pg 5 line 99-100: Fig 1 shows a series of ELISA curves and this sentence states that “high levels of display” were observed. How is this quantified? What is “high” relative to? Can this data be more robustly quantified?

The curves illustrate the binding signal between the anti-gD antibody and the gD tag on the phage surface, dependent on phage concentration. The EC50 value represents the phage concentration where 50% of maximum binding is achieved. A lower EC50 indicates higher levels of gD tag presentation on each phage particle, implying a higher "display level". This results in an inverse correlation between EC50 and "display level". However, due to limitations in accurately measuring phage concentration with OD268, the EC50 value is only semi-quantitative. For reference, an OD268 of 1 typically corresponds to a phage concentration of 5 x 1012 pfu/ml. 

“as demonstrated by low EC50 Values from phage ELISA titration curve” were added in line 103 to clarify the data. 

Pg 5 line 102: State “confirming functional folding” however the signal seems much lower here for a given Phage OD (again it would be useful to have actual quantification of this). Can you comment on this? Is it because a large portion of the expressed peptide is not folded correctly?

As explained in the above question, it’s very hard to accurately determine the exact number of ligands displayed on phage surface, therefore it’s impossible to quantify the fraction of correctly folded peptides displaying on phage. 

The subdued ELISA signal seen for trypsin binding to the DCP ligand displayed on the phage, when compared to the anti-gD and gD tag, could be due to several factors. Firstly, as mentioned by the reviewer, a significant portion of the displayed peptide might not fold correctly. Secondly, the binding between plate-coated trypsin and the phage-displayed peptide might be weak. This could be attributed to the coating of trypsin on a Maxisorp plate potentially distorting the protein's conformation, leading to a weakened binding signal. This occurrence is not uncommon, especially with proteases exhibiting high plasticity. Therefore, the data presented here is more qualitative rather than quantitative.

Pg 13 lines 239-243: Two residues are highlighted based on screening as showing favourable substitutions. In the improved peptide a third substitution has also been made W22Y. Please elaborate further. Were more peptides synthesised and tested and the one currently presented in table 3 is just the best? If so, all of this data should be included.

In fact, additional peptides had been synthesized and the affinity evaluated. The best one is listed in the Table 3. The additional data are listed in supplement Table 2. Based on S2 Table, I24K/R does not improve the affinity. The favorable mutation on this position as indicated by ES heat map could attribute to better display level. In addition, based on this table, W/Y/F at Pos22 are equally tolerated for binding. W7R is the critical change to improve the affinity. We added a brief description on this set of data in line 260-263, where the S2 table is cited. 

Pg 18 line 322-323: I’m not sure about this comment. Many PPIs are much tighter than µM. Please provide a reference to back this up.

Tight PPI usually involve pocket or grooves. For shallow and flat surfaces, according to the new reference added in line 346, PPI with area over ~500A2 is usually needed to achieve stable complex. 

Pg 21 line 394-5: I think it is too strong a claim to say this is a robust platform for discovery of peptide “therapeutics”. Also be what measure is this platform comparable to synthetic antibody libraries? I would say that the peptides being isolated are somewhat weaker binders than things derived from antibody libraries.

In line 417-418, we have toned down the claim to “we have established a robust platform with potential to identify peptide therapeutical if combined with proper strategy of affinity maturation using DCPs as scaffolds.” 

The measure of a library effectiveness is rather arbitrary. Fellouse et. al. (reference cited in the MS) used “hit rate” as a rough indicator. In this paper, they panned a naïve synthetic antibody library against a panel of random proteins and got binders for 12 out of 13 proteins, that is, with hit rate of over 90%. We did the similar arbitrary evaluation on our library against 16 random proteins and observed hit rate of >90% as well. The affinity for primary binder is not the major indicator of the effectiveness for a library. By combining proper strategy of affinity maturation, there’s high chance to obtain high affinity lead eventually. However, due to the primary hits are usually with low affinity, the process for affinity maturation would take much longer time comparing to hit-to-lead process with antibody drugs. 

We changed “high-affinity” to “moderate-affinity” in line 421, and added “However, the process of hit-to-lead for DCP would take much longer time than that for antibody drugs“ in line 422-423 to emphasize this shortcoming. 

Pg 23 line 442: method for phage ELISA should be included since it is used substantially in the main figures. Please also explain how the S/N is calculated

Phage ELISA method has been added in Method section Line 492 -502, where the calculation of S/N is explained. 

Pg 24 peptide synthesis: this method should be included since there are many variations and the folding is non standard. All LC-MS data should be included in the SI

Peptide synthesis method had been added in Method section (Line 504 - 510). All LC-MS data is now added in Supporting Information as S4, which is a zip file containing 51 CoA as pdf files. CoA contains all detailed information, including LC-MS data for each synthesized peptide reported in Table 2, Table 3 and S2 Table in this manuscript. 

Examples of data that should be included in SI:

- Please include all ELISA data. E.g. Pg 9 line 178-180 

A master table that includes all sequences and corresponding ELISA results is now added to Supporting Information as an excel workbook with 16 spreadsheets corresponding to 16 targets. The table is cited as S1 table at line 179 in the main manuscript text. 

After a careful proof-reading Table 1 in line with S1 Table, we realized an error in original Table 1. The total hits for CD16a from EETI-II library should be 44 instead of 57 in previous version. We have made the correction and also correct the number in text in Line 192 accordingly. 

- Please include all peptide LC-MS

Certificates of Analysis (CoA) for 51 peptides reported in Table 2, Table 3 and S2 Table are added in Supporting Information as a zip file containing 51 pdf files with file name the same as the “DCP name” in the tables. This information was cited in the main text in line 212-213 and line 248 as “S4 CoA”.

- Please include all primers used for e.g. library construction

Primers used for library construction are listed in the S3 Table and is cited in line 476 of the main text. 

- Please include full sequences for phage libraries

The full sequences of phagemids for p3 and p8 display, pS2202d and pS2202b, are included in Supporting Information as 2 gb files. They are cited in main text in line 464 as S5 and S6. 

 

Reviewer #2: In this manuscript Zhang et al describe a phage display platform for discovery of peptide therapeutics usingdiverse disulfide-constrained peptides as scaffolds. The screening platform yielded functional hits against a range ofprotein targets including HtrA1 and Notch2. A thorough and cleverly designed affinity maturation strategy is alsodescribed, which enabled hit-to-lead optimisation of a Notch2 ligand, providing a nanomolar binder. The platform could behighly beneficial for discovering therapeutic peptides and thus this paper will be of interest to the peptide community. I amin favour of publication after the minor corrections listed below are addressed.

Minor Corrections

1. Can you comment on the specificity of any of your hits? For example, EETI-II is a known trypsin inhibitor. Do your EETI-II-based HtrA1 inhibitors also inhibit trypsin? As you are optimising for affinity, it would be good to know that you aren’t isolating promiscuous ligands.

We did evaluate the inhibition specificity of HtrA1_EET76 vs. EETI-II against bovine trypsin. The results are added as Fig 3C and is cited in the main text line 233-236. Figure legend for Figure 3 was modified accordingly. As expected, HtrA1_EET76 is a specific inhibitor to HtrA1 and does not inhibit trypsin, in contrasting to the parent scaffold EETI-II that can fully inhibit trypsin activity. 

2. I’m sure this will be fixed in the final manuscript, but I found several figures, in particular figure 2, difficult to read due to poor resolution. Please provide clearer figures.

A new set of figures with higher resolution are re-submitted alone with the revision.

3. Could you include a sentence in the manuscript describing hard and soft randomisation. I think this would be useful for readers not familiar with this concept.

The explanation is added in line 249-252. 

Yours truly,

Yingnan Zhang, Ph.D. and Rami N. Hannoush, Ph.D.

---

## [Decision Letter · Decision Letter 1]

20 Feb 2024

PONE-D-23-32053R1Disulfide-constrained peptide scaffolds enable a robust peptide-therapeutic discovery platformPLOS ONE

Dear Dr. Zhang,

Thank you for submitting your revised manuscript to PLOS ONE. Before its formal acceptance, reviewer 1 indicated the need of a minor but relevant acclaration in the introduction. Please submit your revised manuscript by Apr 05 2024 11:59PM. If you will need more time than this to complete your revisions, please reply to this message or contact the journal office at plosone@plos.org. Please include the following items when submitting your revised manuscript:A rebuttal letter that responds to each point raised by the academic editor and reviewer(s). You should upload this letter as a separate file labeled 'Response to Reviewers'.A marked-up copy of your manuscript that highlights changes made to the original version. You should upload this as a separate file labeled 'Revised Manuscript with Track Changes'.An unmarked version of your revised paper without tracked changes. You should upload this as a separate file labeled 'Manuscript'.If applicable, we recommend that you deposit your laboratory protocols in protocols.io to enhance the reproducibility of your results. Protocols.io assigns your protocol its own identifier (DOI) so that it can be cited independently in the future. For instructions see: https://journals.plos.org/plosone/s/submission-guidelines#loc-laboratory-protocols. Additionally, PLOS ONE offers an option for publishing peer-reviewed Lab Protocol articles, which describe protocols hosted on protocols.io. Read more information on sharing protocols at https://plos.org/protocols?utm_medium=editorial-email&utm_source=authorletters&utm_campaign=protocols.

We look forward to receiving your revised manuscript.

Kind regards,

Maria Gasset, Ph.D.

Academic Editor

PLOS ONE

Journal Requirements:

Reviewers' comments:

Reviewer's Responses to Questions

**Comments to the Author**

1. If the authors have adequately addressed your comments raised in a previous round of review and you feel that this manuscript is now acceptable for publication, you may indicate that here to bypass the “Comments to the Author” section, enter your conflict of interest statement in the “Confidential to Editor” section, and submit your "Accept" recommendation.

Reviewer #1: (No Response)

Reviewer #2: All comments have been addressed

2. Is the manuscript technically sound, and do the data support the conclusions?

Reviewer #1: Yes

Reviewer #2: Yes

3. Has the statistical analysis been performed appropriately and rigorously? 

Reviewer #1: Yes

Reviewer #2: Yes

4. Have the authors made all data underlying the findings in their manuscript fully available?

Reviewer #1: Yes

Reviewer #2: Yes

5. Is the manuscript presented in an intelligible fashion and written in standard English?

Reviewer #1: Yes

Reviewer #2: Yes

6. Review Comments to the Author

Reviewer #1: On the whole I am happy that the authors have addressed mine and the other reviewers comments. However I disagree with one response:

Original reviewing comment: - similar library approaches have been applied with mRNA display e.g. Liu et al

JACS 143 18481 2021 and this work should be mentioned and cited in the

introduction.

Response: Add “peptide library approaches have been applied with mRNA display” at line 44 and

the reference is included.

Further comment: This sentence has been added before the comments on DCPs as if it refers only to general mRNA display screens. My point was rather that mRNA display has been used to create modified DCP peptides in a similar manner (using a cyclotide MCoTI-II scaffold) to the screens in this paper (as shown in the cited paper) and this is what I think should be made clear in the introduction.

Reviewer #2: (No Response)

7. PLOS authors have the option to publish the peer review history of their article (what does this mean?). If published, this will include your full peer review and any attached files.

Reviewer #1: No

Reviewer #2: **Yes: **Scott Lovell

---

## [Author Response · Author response to Decision Letter 1]

20 Feb 2024

Dear Editor:

Thank you for your careful review of our revised manuscript titled “Disulfide-constrained peptide scaffolds enable a robust peptide-therapeutic discovery platform” (PONE-D-23-32053R1). We appreciated the positive feedbacks and comments from both reviewers. Following are our response to one point raised by Reviewer1. The reviewer’s comment is in bold font and our response are in regular font:

Reviewer #1: On the whole I am happy that the authors have addressed mine and the other reviewers comments. However I disagree with one response:

Original reviewing comment: - similar library approaches have been applied with mRNA display e.g. Liu et al

JACS 143 18481 2021 and this work should be mentioned and cited in the

introduction.

Response: Add “peptide library approaches have been applied with mRNA display” at line 44 and

the reference is included.

Further comment: This sentence has been added before the comments on DCPs as if it refers only to general mRNA display screens. My point was rather that mRNA display has been used to create modified DCP peptides in a similar manner (using a cyclotide MCoTI-II scaffold) to the screens in this paper (as shown in the cited paper) and this is what I think should be made clear in the introduction.

We have modified the Introduction to clarify that mRNA display had been successfully applied to DCP library strategies and our phage display approach has extended the applicability of the strategy. From line 43-49, the new text is:

“To overcome some of these peptide-intrinsic shortcomings, we considered using naturally occurring and highly stable disulfide constrained peptides (DCPs) as frameworks coupled with display technology aimed at generating de novo binders of therapeutically relevant target proteins [3]. Notebly, DCP library strategies has been effectively employed using mRNA display with MCoTI-II as a scaffold [4]. In this study, we employed phage display across seven distinct scaffolds to extent the applicability of this approach. “

Yours truly,

Yingnan Zhang, Ph.D. and Rami N. Hannoush, Ph.D.

---

## [Editor Report · Decision Letter 2]

23 Feb 2024

Disulfide-constrained peptide scaffolds enable a robust peptide-therapeutic discovery platform

PONE-D-23-32053R2

Dear Dr. Yingnan Zhang,

We’re pleased to inform you that your manuscript has been judged scientifically suitable for publication and will be formally accepted for publication once it meets all outstanding technical requirements.

Kind regards,

Maria Gasset, Ph.D.

Academic Editor

PLOS ONE
---

## [Editor Report · Acceptance letter]

12 Mar 2024

PONE-D-23-32053R2 

PLOS ONE

Dear Dr. Zhang, 

I'm pleased to inform you that your manuscript has been deemed suitable for publication in PLOS ONE. Congratulations! Your manuscript is now being handed over to our production team.

Kind regards, 

on behalf of

Dr. Maria Gasset 

Academic Editor

PLOS ONE